# Multi-Criteria Selection of Electric Delivery Vehicles Using Fuzzy–Rough Methods

**Ning Wang [1]**, **Yong Xu [1,\***, **Adis Puška [2]**, **Željko Stević [3,\***** and **Adel Fahad Alrasheedi [4]**

1    School of Management, Shandong Technology and Business University, No. 191 Binhai Zhong Road, Laishan District, Yantai 264005, China; wangning114@sdtbu.edu.cn

2    Department of Public Safety, Government of Brčko District of Bosnia and Herzegovina, Bulevara Mira 1, 76100 Brčko, Bosnia and Herzegovina; adispuska@yahoo.com

3    Faculty of Transport and Traffic Engineering, University of East Sarajevo, Vojvode Mišića 52, 74000 Doboj, Bosnia and Herzegovina

4    Department of Statistics and Operations Research, College of Science, King Saud University, P.O. Box 2455, Riyadh 11451, Saudi Arabia; aalrasheedi@ksu.edu.sa

\*    Correspondence: 201613583@sdtbu.edu.cn (Y.X.); zeljkostevic88@yahoo.com or zeljko.stevic@sf.ues.rs.ba (Ž.S.)

**Abstract:** Urban logistics implementation causes environmental pollution; therefore, it is necessary to consider the impact on the environment when carrying out such logistics. Electric vehicles are alternative vehicles that reduce the impact on the environment. For this reason, this study investigated which electric vehicle has the best indicators for urban logistics. An innovative approach when selecting such vehicles is the application of a fuzzy–rough method based on expert decision making, whereby the decision-making process is adapted to the decision makers. In this case, two methods of multi-criteria decision making (MCDM) were used: SWARA (stepwise weight assessment ratio analysis) and MARCOS (measurement alternatives and ranking according to compromise solution). By applying the fuzzy–rough approach, uncertainty is included when making a decision, and it is possible to use linguistic values. The results obtained by the fuzzy–rough SWARA method showed that the range and price of electric vehicles have the greatest influence on the selection of an electric delivery vehicle. The results of applying the fuzzy–rough MARCOS method indicated that the Kangoo E-Tech Electric vehicle has the best characteristics according to experts' estimates. These results were confirmed by validation and the application of sensitivity analysis. In urban logistics, the selection of an electric delivery vehicle helps to reduce the impact on the environment. By applying the fuzzy–rough approach, the decision-making problem is adjusted to the preferences of the decision makers who play a major role in purchasing a vehicle.

**Keywords:** selection of electric delivery vehicles; urban logistics; sustainability; fuzzy–rough numbers





## 1. Introduction

The increasing population in cities has made the problem of distributing goods to buyers and sellers more complex. In these areas, due to the expansion of online shopping, there is an increased demand for goods [1,2], so more and more goods need to be distributed. This expansion in online shopping has occurred not only in urban areas, but also in all other areas [3], so the amount of goods distributed in rural areas is also significant. Last-mile logistics prevails in this type of distribution. This area of logistics includes all logistics activities related to the delivery of goods to final customers in urban areas [4,5]. Last-mile delivery in urban logistics is the most complex and can cause up to 28% of all logistics costs [6]. Due to the specificity of urban areas and the increasing distribution of goods in such locations, the environment is negatively affected [7,8]. This type of logistics has a significant impact on air pollution [9], so the application of sustainable urban logistics is increasingly being imposed. For this reason, the European Commission requires the countries of the European Union to switch to environmentally friendly transportation [10].

This is because urbanization processes lead to a constant increase in the number of people and the amount of goods distributed, which has negative impacts on the environment and the quality of life in urban areas [11,12].

Sustainable urban logistics refers to the organization of the efficient transportation [13], distribution [14], and delivery of goods with minimal impact on the environment and society. The problems with the implementation of this type of logistics arise from the fact that it involves different interested parties and there are restrictions dictated by the urban environment and the position of the customer that affect last-mile delivery [15]. In addition, the challenges of sustainable urban logistics include increased volumes of goods for distribution, the burden of urban areas with an increasing number of delivery vehicles, the increased costs of distribution in urban areas, the difficulties of on-time delivery, and an aging workforce [4]. Hence, in practice, the possibility of introducing sustainable means of transport into urban logistics is targeted. The reason for this is that the use of road transportation in urban logistics emits three to five times more $CO_2$ than other means of transportation [16]. In practice, a solution is being sought, and so new approaches that attempt to apply sustainability in urban logistics are being introduced. Currently, the best solutions are the use of electric vehicles and hydrogen vehicles [17]. Due to the small number of hydrogen-powered vehicles, the selection of an electric delivery vehicle for distribution in urban areas is the focus of this paper [10]. However, electric delivery vehicles have certain limitations that have slowed down their introduction. The main limitations are related to the range, battery capacity, charging time, and price of these vehicles. Due to these limitations, decision makers in companies still opt for classic delivery vehicles. However, electric vehicles do not emit harmful gases into the atmosphere, unlike classic internal combustion vehicles [9]. The use of electric vehicles in urban logistics would reduce the impact on the environment. In order to reduce this impact, the energy used for charging electric vehicles must be produced from sustainable energy sources. Based on this, the motivation for this paper was to apply sustainability in urban logistics using electric delivery vehicles.

This paper is intended to promote the use of electric vehicles in the distribution of goods using examples from practice. The aim of this study was to select an electric delivery vehicle that would best meet the sustainability goals of urban logistics for the distribution of goods. Electric delivery vehicles have similar technical characteristics, and the selection of these vehicles cannot be based only on these technical features. In this research, a combined fuzzy–rough approach was applied in the selection of electric delivery vehicles based on fuzzy–rough SWARA and MARCOS methods. When applying the fuzzy–rough approach in decision making, specifically in the selection of electric delivery vehicles, the entire decision-making process is adjusted to the users' requirements. Thus, the decision is not only influenced by the technical characteristics of these vehicles, but also includes the users' preferences, which is achieved by applying this approach. In addition, the combination of fuzzy and rough approaches first enables linguistic values to be used in decision making, which is more suitable for decision makers. Then, the fuzzy approach enables these linguistic values to be used to obtain final results, while the application of the rough approach in this decision making includes uncertainty in the decision-making process. By applying the fuzzy–rough approach, all the advantages of both fuzzy and rough approaches are exploited, and thus the two approaches complement each other. That is why this approach is better than classic fuzzy and rough approaches in decision making [18].

Based on this, the contributions of this paper are reflected in the following:

- Improving sustainable urban logistics using electric delivery vehicles.
- Applying an innovative methodology for the selection of an electric delivery vehicle based on a fuzzy–rough approach.
- Using a fuzzy–rough approach when selecting electric delivery vehicles, adapting the decision-making process to human preferences.
- Selecting an electric vehicle that best meets the sustainability goals of urban logistics.

- Promoting the use of electric delivery vehicles in urban logistics applications.

Apart from the introduction, this paper is divided into the following selections. Section 2 will present the background of the research. Section 3 will provide the characteristics of the fuzzy–rough approach and describe the methods. In Section 4, the case study of this research will be presented. Section 5 will include the selection of electric delivery vehicles and additional analyses, including a sensitivity analysis of the criteria and the validation of the results. Section 6 will contain a discussion on the obtained research results. In Section 7, concluding considerations will be provided, and the limitations of the research and guidelines for future research will be presented.

## 2. Background of the Research

### 2.1. Sustainable Urban Logistics

In order to apply sustainable urban logistics, it is necessary to select vehicles that will help in carrying out such logistics. In previous papers, many authors have highlighted the fact that it is necessary to use vehicles that help achieve sustainable urban logistics. In their paper, Duarte et al. [19] investigated the possibility of using alternative vehicles such as battery electric vehicles. They proved that these vehicles reduce energy consumption and can be used in urban logistics. Jones et al. [20] considered the application of hydrogen-powered vehicles as an alternative in freight transportation in urban areas. They showed that these vehicles are economically competitive and that it is necessary to apply them in sustainable urban logistics. In their paper, Schöder et al. [21] considered the use of new technologies in urban logistics and determined that the application of electric vehicles is still an underestimated element for the application of sustainable urban logistics.

Oliveira et al. [22] conducted a systematic review of the literature and confirmed that in urban areas, especially when it comes to last-mile distribution, it is necessary to use bicycles, tricycles, and light commercial electric vehicles. In their research, Napoli et al. [11] dealt with the application of electric vehicles in the distribution of cargo using an example from Sicily. They investigated the application of these vehicles because some cities in the European Union (EU) have begun to implement measures to support low or zero emissions, increasing the use of alternative fuels or electric vehicles. Navarro et al. [23] analyzed smart urban logistics in their paper and studied the combined application of electric tricycles in sustainable urban logistics. In their paper, Melo and Baptista [24] considered the application of cargo bikes over short distances in urban settlements. They determined that these cargo bikes can replace 10% of conventional delivery vans, thus reducing pollution and contributing to sustainable urban logistics.

Settey et al. [25] dealt with the problem of sustainable urban logistics and analyzed the use of electric vehicles. However, they determined that the use of these vehicles over longer distances is questionable due to the distance between distribution centers, so they suggested the use of hybrid goods vehicles in urban logistics. Bac and Erdem [26] considered the possibility of using electric vehicles in urban logistics. Due to the limitations of these vehicles, i.e., the range and charging time, and the insufficient number of fast charging stations, the movement routes of these vehicles must be optimized in order to apply them in sustainable urban logistics in the best way.

From these and other similar papers, it can be concluded that different vehicles can be used when applying sustainable urban logistics. Thus, small delivery vehicles such as electric bicycles and tricycles can be used for shorter distances, while electric delivery vehicles can be used for medium distances. Therefore, when applying sustainable urban logistics, different vehicles should be used in order to optimize the process.

### 2.2. Application of Multi-Criteria Methods in the Selection of Electric Vehicles for Sustainable Logistics

Many factors influence the selection of a vehicle in sustainable urban logistics. In the decision-making process, these factors are presented as criteria by which alternatives are considered. In order to solve decision-making problems, MCDM methods are used

in practice. Further, this paper will provide a review of research papers in which these methods have been used in the selection of electric vehicles.

In his paper, Ziemba [27] selected electric vehicles for the Polish market using the NEAT F-PROMETHEE (New Easy Approach To Fuzzy PROMETHEE) method and combined it with Monte Carlo simulation and elements of the SMAA (Stochastic Multi-criteria Acceptability Analysis) method. Więckowski et al. [28] used the TOPSIS (Technique for Order of Preference by Similarity to Ideal Solution) method and interval numbers to evaluate electric cars in order to achieve sustainability in transportation. Štilić et al. [29] used the SWARA method, MSDM (Modified Standard Deviation Method), and MABAC (Multi-Attributive Border Approximation Area Comparison) method in order to select an electric car that best met sustainability goals in urban areas using the example of taxi services in the Brčko District of Bosnia and Herzegovina.

In his paper, Ziemba [30] indicated the possibility of using electric cars to reduce greenhouse gas emissions and meet sustainability goals. In addition, he selected electric cars using the PROSA-C (PROMETHEE for Sustainability Assessment-Criteria) method combined with Monte Carlo simulation. Anastasiadou et al. [31] emphasized the need for sustainable planning in transportation and offered autonomous electric vehicles as a solution. For this purpose, they used the AHP, TOPSIS, and VIKOR (*Višekriterijumsko kompromisno rangiranje*, in Serbian) methods in the analysis. Onat et al. [32] investigated sustainability improvements for passenger cars and evaluated the impact of different vehicles on sustainable transportation, using a multi-objective optimization model. Wei and Zhou [33] pointed out that electric vehicles show potential in reducing $CO_2$ emissions and that these vehicles must be used by government agencies and public bodies. In order to purchase these vehicles, they selected sustainable suppliers using the BWM (best-worst method) and VIKOR method.

Yavuz et al. [34] considered the problem of selecting vehicles running on alternative fuels for fleet operations. For this purpose, they used the multi-criteria HFLTS (Hesitant Fuzzy Linguistic Term Sets) method and found that electric vehicles showed the best applicability in practice. Kijewska et al. [35] drew attention to the problem of pollution in urban areas and offered a solution to the problem by using electric cars. In their paper, they selected the electric car that best met the set goals using Promethee II (protracted method that requires a priori knowledge of the criteria weights II). Puška et al. [36] revealed the possibility of introducing electric vehicles to reduce the impact on the environment. They selected electric vehicles in order to achieve this goal, using the DNMEREC (Double Normalization Method based on the Removal Effects of Criteria) and DNCRADIS (Double Normalization Compromise Ranking of Alternatives from Distance to Ideal Solution) methods.

In their paper, Tian et al. [37] found that electric vehicles are one of the most popular forms of low-carbon transportation and help preserve the environment. Through the BWM and ORESTE (*Organísation, rangement et Synthése de données relarionnelles*, in French) method, they provided support for purchasing electric vehicles. Aboushaqrah et al. [38] selected alternative-fuel taxis in order to make sustainable transportation more efficient. For this purpose, they used the AHP (Analytic Hierarchy Process), Neutrosophic sets, and TOPSIS methods. Biswas et al. [39] addressed the possibility of reducing the impact on the environment and applying sustainability principles using electric vehicles. In their paper, they used the FUCOM (Full Consistency Method) and AROMAN (Alternative Ranking Order Method with Two-Step Normalization) approaches. Based on these papers, it can be seen that many authors have identified the possibility of using electric vehicles in urban areas in order to reduce the impact on environmental pollution. This is why electric vehicles are currently the best alternative to vehicles with internal combustion engines, so it is necessary to consider the possibility of using these vehicles in urban logistics. In addition, a challenge in selecting electric delivery vehicles is that the observed characteristics of the vehicles are contradictory, so it is necessary to find a compromise between these specifications. It is not possible to find a vehicle that has the lowest price; the largest range and battery (which, at the same time, can be charged quickly); and a large

load capacity and space for placing goods during delivery. Therefore, in the above research papers, as well as in the current study, the aim was to find an electric delivery vehicle that best met the requirements of the vehicle's user.

### 2.3. Research Gaps

This research addressed several research gaps. Various methods and procedures were used for the selection of electric vehicles in the abovementioned research papers. These selections were based mainly on the technical characteristics of vehicles [27–30,35,36], though linguistic evaluations [27,29–31,33] and objective methods [35] were also used to determine the importance of criteria, while some papers used a combination of technical characteristics and expert linguistic assessments [39]. However, the fuzzy–rough approach has not been applied in the selection of such vehicles until now. This approach was used because the technical specificities of certain electric vehicles for urban logistics are similar, and there is no significant difference. Thus, it is possible for one criterion to change the final ranking and cause certain electric vehicles to be ranked better or worse, especially when conducting sensitivity analyses to examine the importance of a certain criterion in the ranking of alternatives [29]. By applying the fuzzy–rough approach, decision making is adapted to the decision makers, who use subjective estimates to evaluate the vehicles and, based on these estimates, make a choice.

The selected fuzzy–rough approach first brings the decision-making process closer to human thinking by applying the fuzzy approach, while by applying the rough approach it reduces subjectivity in decision making and introduces uncertainty. In this way, a more comprehensive decision is made, and the decision is not based solely on technical characteristics, but also on subjective evaluations. If in practice only the technical specifications of vehicles were considered, most users would have the same vehicle, which is not the case. This is because the selection of vehicles is a typical example where personal preferences influence the final decision. Among a huge number of available vehicles, decision makers choose to buy a specific vehicle and make that decision based on personal preferences.

Many methods have been used in the fuzzy–rough approach, but the fuzzy–rough MARCOS method has not been used in research up to now, even though the MARCOS method has been used in several hundred papers. Thus, we will show how certain MCDM methods can be further developed, with the fuzzy–rough approach being one of these methods. Based on this, this paper provides guidelines for developing the method and its application in MCDM problems. The methodology of using subjective ratings in the selection of electric vehicles is not new, since it was used in the paper of Biswas et al. [39], but the use of the fuzzy–rough approach for such a selection is novel and represents a shift in practical applications. This approach opens up the possibility of further upgrading the decision-making process, so that it is possible to combine other approaches in future research.

## 3. Methodology and Methods

During the implementation of this research, the phases shown in Figure 1 were used.

When conducting this research, first, experts were selected, who then identified the limitations in the selection of alternatives and the criteria by which these alternatives would be assessed. After this, the experts evaluated the criteria and alternatives using linguistic values. In order to use these values when obtaining results, the values were first transformed into fuzzy numbers, and then the lower and upper limits of individual fuzzy numbers were determined and an initial decision matrix was formed. This matrix represented the basis for conducting the analyses, i.e., for determining weights using the fuzzy–rough SWARA method and forming a ranking list using the fuzzy–rough MARCOS method. Finally, the obtained results were confirmed by validating the results and determining the importance of the criteria, forming a ranking list of alternatives via a sensitivity analysis.

| | |
|---|---|
| Phase 1. Preparation of research phases<br>- Selection of experts<br>- Identification of limitations when<br>selecting alternatives<br>- Selection of criteria | Phase 2. Evaluation of criteria and<br>alternatives<br>- Evaluation of criteria by experts<br>- Evaluation of alternatives by experts |
| Phase 3. Preparation for analysis<br>- Transformation of linguistic estimates<br>into fuzzy numbers<br>- Application of rough approach<br>- Formation of an initial decision matrix | Phase 4. Conducting the analysis<br>- Determination of weights using the<br>fuzzy-rough SWARA method<br>- Ranking of alternatives using the fuzz-<br>rough MARCOS method |

Phase 5. Confirmation of results
- Carrying out the validation of the
results
- Carrying out sensitivity analysis

**Figure 1.** Research phases.

The purpose of this methodology is to bring decision making closer to the decision maker. The decision maker does not have to provide exact information, but it is possible to use imprecise information in the form of linguistic values [40]. These values make it possible to evaluate not only qualitative but also quantitative criteria [39]. In this case, estimates are given in the form of good, bad, medium, etc. It is easier for people to say that something is good, very good, or excellent than to evaluate it with numerical ratings using, e.g., a grade of 4 or 5 [41]. By using these values, this research is more adapted to human thinking.

### 3.1. Preliminaries

In practice, there are different ways to make decisions. Thus, a fuzzy set that uses linguistic values was first formed [40] in order to employ imprecise information in decision making. Linguistic values are used in many studies and have shown their advantages and applicability [41,42]. Subsequently, a rough set enabling decision making in conditions of uncertainty with a reduced subjective influence from decision makers was created [43]. In addition, the rough set proved to be effective when data are imprecise and vague and when it is necessary to make a decision with such information. Combining these two approaches, we attempted to exploit the advantages of both, thus enabling the use of fuzzy numbers with lower and upper limits defined by the rough approach. Fuzzy numbers are used as crisp numbers in the rough approach.

In this approach, the fuzzy–rough numbers share the concepts of lower and upper limits and rough boundary intervals with classic rough numbers [18]. The novelty of this approach is that individual fuzzy numbers are viewed as crisp numbers and rough set operations are performed on them. Thus, linguistic values are first transformed into fuzzy numbers using the membership function, and then individual fuzzy numbers are observed in order to determine their lower and upper limits using the rough approach.

Assume that the universe $U$ contains all risk estimates, while $Y$ is an arbitrary object of the universe $U$. $\theta^e = (\widetilde{X}_1, \widetilde{X}_2, \ldots \widetilde{X}_n)$ is a cluster that covers all elements in the universe $U$, and if the values of $\widetilde{X}_i$ are fuzzy values, then $\widetilde{X}_i = \left(x_i^l, x_i^m, x_i^u\right)(i = 1, 2, \ldots, n)$. This is actually a fuzzy risk estimate denoted as a triangular fuzzy number. If we assume that

$\theta^e = \{x_1^e, x_2^e, \ldots, x_n^e\}$ $(e = l, m, n)$, then the lower and upper limits of the element $\overset{\sim}{X}_i$ can be defined as follows [44]:

$$\underline{Lim}(c_i^e) = \frac{1}{N^e} \sum_{i=1}^{\underline{N^e}} \varphi \epsilon \underline{Apr}(c_i^e), \tag{1}$$

$$\overline{Lim}(c_i^e) = \frac{1}{N^e} \sum_{i=1}^{\overline{N^e}} \varphi \epsilon \overline{Apr}(c_i^e) \tag{2}$$

When the lower and upper limits of the triangular fuzzy numbers are determined, the fuzzy–rough number $\overset{\sim}{X}_i$ can be represented as follows [45]:

$$FR\left(\overset{\sim}{X}_i\right) = \left(\left[x_i^{lL}, x_i^{lU}\right], \left[x_i^{mL}, x_i^{mU}\right], \left[x_i^{uL}, x_i^{uU}\right]\right) = \left(\left[\underline{Lim}\left(x_i^l\right), \overline{Lim}\left(x_i^l\right)\right], \left[\underline{Lim}(x_i^m), \overline{Lim}(x_i^m)\right], \left[\underline{Lim}(x_i^u), \overline{Lim}(x_i^u)\right]\right) \tag{3}$$

If there are two fuzzy rough numbers $FR(\overline{a}) = \left(\left[a^{lL}, a^{lU}\right], \left[a^{mL} a\alpha^{mU}\right], \left[a^{uL}, a^{uU}\right]\right)$ and $FR\left(\overline{b}\right) = \left(\left[b^{lL}, b^{lU}\right], \left[b^{mL}, b^{mU}\right], \left[b^{uL}, b^{uU}\right]\right)$, the operations performed on them are as follows:

Addition:

$$\begin{aligned} FR(\overline{a}) + FR\left(\overline{b}\right) &= \left(\left[a^{lL}, a^{lU}\right], \left[a^{mL}, a^{mU}\right], \left[a^{uL}, a^{uU}\right]\right) + \left(\left[b^{lL}, b^{lU}\right], \left[b^{mL}, b^{mU}\right], \left[b^{uL}, b^{uU}\right]\right) \\ &= \left[a^{lL} + b^{lL}, a^{lU} + b^{lU}\right], \left[a^{mL} + b^{mL}, a^{mU} + b^{mU}\right], \left[a^{uL} + b^{uL}, a^{uU} + b^{uU}\right] \end{aligned} \tag{4}$$

Subtraction:

$$\begin{aligned} FR(\overline{a}) - FR\left(\overline{b}\right) &= \left(\left[a^{lL}, a^{lU}\right], \left[a^{mL}, a^{mU}\right], \left[a^{uL}, a^{uU}\right]\right) + \left(\left[b^{lL}, b^{lU}\right], \left[b^{mL}, b^{mU}\right], \left[b^{uL}, b^{uU}\right]\right) \\ &= \left[a^{lL} - b^{uU}, a^{lU} - b^{uL}\right], \left[a^{mL} - b^{mU}, a^{mU} - b^{mL}\right], \left[a^{uL} - b^{lU}, a^{uU} - b^{lL}\right] \end{aligned} \tag{5}$$

Multiplication:

$$\begin{aligned} FR(\overline{a}) \times FR\left(\overline{b}\right) &= \left(\left[a^{lL}, a^{lU}\right], \left[a^{mL}, a^{mU}\right], \left[a^{uL}, a^{uU}\right]\right) + \left(\left[b^{lL}, b^{lU}\right], \left[b^{mL}, b^{mU}\right], \left[b^{uL}, b^{uU}\right]\right) \\ &= \left[a^{lL} \times b^{lL}, a^{lU} \times b^{lU}\right], \left[a^{mL} \times b^{mL}, a^{mU} \times b^{mU}\right], \left[a^{uL} \times b^{uL}, a^{uU} \times b^{uU}\right] \end{aligned} \tag{6}$$

Division:

$$\begin{aligned} FR(\overline{a}) \div FR\left(\overline{b}\right) &= \left(\left[a^{lL}, a^{lU}\right], \left[a^{mL}, a^{mU}\right], \left[a^{uL}, a^{uU}\right]\right) + \left(\left[b^{lL}, b^{lU}\right], \left[b^{mL}, b^{mU}\right], \left[b^{uL}, b^{uU}\right]\right) \\ &= \left[a^{lL} \div b^{uU}, a^{lU} \div b^{uL}\right], \left[a^{mL} \div b^{mU}, a^{mU} \div b^{mL}\right], \left[a^{uL} \div b^{lU}, a^{uU} \div b^{lL}\right] \end{aligned} \tag{7}$$

Scalar multiplication:

$$c \times FR(\overline{a}) = c \times \left(\left[a^{lL}, a^{lU}\right], \left[a^{mL}, a^{mU}\right], \left[a^{uL}, \alpha^{uU}\right]\right) = \left(\left[c \times a^{lL}, c \times a^{lU}\right], \left[c \times a^{mL}, c \times a^{mU}\right], \left[c \times a^{uL}, c \times a^{uU}\right]\right) \tag{8}$$

Scalar division:

$$\frac{RF(\overline{a})}{c} = \frac{\left[a^{lL}, a^{lU}\right], \left[a^{mL}, a^{mU}\right], \left[a^{uL}, \alpha^{uU}\right]}{c} = \left(\left[\frac{a^{lL}}{c}, \frac{a^{lU}}{c}\right], \left[\frac{a^{mL}}{c}, \frac{a^{mU}}{c}\right], \left[\frac{a^{uL}}{c}, \frac{a^{uU}}{c}\right]\right) \tag{9}$$

### 3.2. Fuzzy–Rough SWARA Method

The fuzzy–rough SWARA method was used to determine the importance of particular criteria. In this study, the SWARA method created by Keršulienė et al. [46] was modified. The purpose of this method is to allow decision makers to first rank criteria according to their importance and then determine their weights by comparing the criteria. Whereas in the AHP method each criterion must be compared with each other criterion [47], this does not have to be carried out in the SWARA method. The ranking of the criteria and



their subsequent evaluation based on the ranking are performed. In this way, it is not necessary to compare each individual criterion, but only the worse-ranked compared to the better-ranked. Thus, the number of comparisons is reduced to n − 1 [48] in contrast to AHP, where the number of comparisons is n·(n − 1) [49]. The problem with the application of the AHP method is the number of criteria that are compared [50]. If there are a large number of criteria, it is difficult to achieve consistency in decision making. This is another reason why the SWARA method was used. This method contains the following steps [51]:

Step 1. Defining a group of criteria.

Step 2. Defining a team of experts.

Step 3. Evaluation of the criteria by experts. Experts evaluate these criteria with linguistic ratings. In this step, the experts determine how important each criterion is and assign specific ratings to individual criteria.

Step 4. Transformation of individual expert ratings into a group fuzzy–rough initial decision matrix.

$$FR_i(\overline{X_i}) = \left( \left[ x_i^{lL}, x_i^{lU} \right], \left[ x_i^{mL}, x_i^{mU} \right], \left[ x_i^{uL}, x_i^{uU} \right] \right) \tag{10}$$

Step 5. Ranking the criteria using the initial decision matrix.

Step 6. Determining the importance of the criteria by normalizing the initial decision matrix.

$$FRN(N_j) = \left[ \left( n_j^{L1}, n_j^{U1} \right), \left( n_j^{L2}, n_j^{U2} \right), \left( n_j^{L3}, n_j^{U3} \right) \right]_{1 \times m} \tag{11}$$

The criterion with the greatest importance receives a value of 1 for all elements of the fuzzy–rough number.

$$\left[ \left( n_j^{L1}, n_j^{U1} \right), \left( n_j^{L2}, n_j^{U2} \right), \left( n_j^{L3}, n_j^{U3} \right) \right] = [(1.00, 1.00), (1.00, 1.00), (1.00, 1.00)] \tag{12}$$

The values of the other criteria are determined in relation to the criterion with the highest importance. If the criteria have the same importance, $j = 1$, then only the value of the previous criterion is copied. If the criterion has a lower importance, $j > 1$, then the importance of the criterion is calculated as follows:

$$FRN(N_j) = \left[ \left( \frac{n_j^{L1}}{z_j^{U3}}, \frac{n_j^{U1}}{z_j^{L3}} \right), \left( \frac{n_j^{L2}}{z_j^{U2}}, \frac{n_j^{U2}}{z_j^{L2}} \right), \left( \frac{n_j^{L3}}{z_j^{U1}}, \frac{n_j^{U3}}{z_j^{L1}} \right) \right]_{1 \times m} \quad j = 2, 3, \ldots, m \tag{13}$$

It is crucial that the values of the criterion expressed as an average fuzzy–rough number are ordered by size from the highest to the lowest.

Step 7. Determining the relative importance of criterion $FRN(\Im_j)$.

$$FRN(\Im_j) = \left[ \left( \Im_j^{L1}, \Im_j^{U1} \right), \left( \Im_j^{L2}, \Im_j^{U2} \right), \left( \Im_j^{L3}, \Im_j^{U3} \right) \right]_{1 \times m} \tag{14}$$

This is accomplished using the following equation:

$$FRN(\Im_j) = \left[ \left( n_j^{L1} + 1, n_j^{U1} + 1 \right), \left( n_j^{L2} + 1, n_j^{U2} + 1 \right), \left( n_j^{L3} + 1, n_j^{U3} + 1 \right) \right]_{1 \times m} \quad j = 2, 3, \ldots, m \tag{15}$$

In this step, the normalized value of the fuzzy number is added to the number 1; however, the most important criterion retains all values of 1 of the fuzzy–rough number (Equation (12)).

Step 8. Computation of the matrix of recalculated weights for the criteria $FRN(\Re_j)$.

$$FRN(\Re_j) = \left[ \left( \Re_j^{L1}, \Re_j^{U1} \right), \left( \Re_j^{L2}, \Re_j^{U2} \right), \left( \Re_j^{L3}, \Re_j^{U3} \right) \right]_{1 \times m} \tag{16}$$

The elements of this matrix are formed as follows:

$$
FRN(\Re_j)
\begin{bmatrix}
\Re_j^{L1} = \begin{pmatrix} 1.00 \ j = 1 \\ \dfrac{\Re_{j-1}^{L1}}{\Im_j^{U3}} \ j > 1 \end{pmatrix}, & \Re_j^{U1} = \begin{pmatrix} 1.00 \ j = 1 \\ \dfrac{\Re_{j-1}^{U1}}{\Im_j^{L3}} \ j > 1 \end{pmatrix}, \\[4mm]
\Re_j^{L2} = \begin{pmatrix} 1.00 \ j = 1 \\ \dfrac{\Re_{j-1}^{L2}}{\Im_j^{U2}} \ j > 1 \end{pmatrix}, & \Re_j^{U2} = \begin{pmatrix} 1.00 \ j = 1 \\ \dfrac{\Re_{j-1}^{U2}}{\Im_j^{L2}} \ j > 1 \end{pmatrix}, \\[4mm]
\Re_j^{L3} = \begin{pmatrix} 1.00 \ j = 1 \\ \dfrac{\Re_{j-1}^{L3}}{\Im_j^{U1}} \ j > 1 \end{pmatrix}, & \Re_j^{U3} = \begin{pmatrix} 1.00 \ j = 1 \\ \dfrac{\Re_{j-1}^{U3}}{\Im_j^{L1}} \ j > 1 \end{pmatrix},
\end{bmatrix}
\tag{17}
$$

In cases where two or more criteria have equal importance, the following equation should be applied:

$$
FRN(\Re_j) = FRN(\Re_{j-1})
\tag{18}
$$

Step 9. Computing the final weights of criteria $FRN(W_j)$, which is carried out as follows:

$$
FRN(W_j) = \left[ \frac{FRN(\Re_j)}{FRN(\aleph_j)} \right]
\tag{19}
$$

where $FRN(\aleph_j) = \sum_{j=1}^{m} FRN(\Re_j)$. These final weight values are obtained using the following equation:

$$
FRN(W_j) = \left[ \left( \frac{\Re_j^{L1}}{\aleph_j^{U3}}, \frac{\Re_j^{U1}}{\aleph_j^{L3}} \right), \left( \frac{\Re_j^{L2}}{\aleph_j^{U2}}, \frac{\Re_j^{U2}}{\aleph_j^{L2}} \right), \left( \frac{\Re_j^{L3}}{\aleph_j^{U1}}, \frac{\Re_j^{U3}}{\aleph_j^{L1}} \right) \right]_{1 \times m} \quad j = 2, 3, \ldots, m
\tag{20}
$$

### 3.3. Fuzzy–Rough MARCOS Method

When applying the fuzzy–rough method, a modification of the classic MARCOS method created by Stević et al. [52] was performed. This is a recent method for multi-criteria analysis and, as such, has been used in many studies in a short period of time and has been accepted by various authors. Furthermore, its results do not deviate from the results of other methods and have been confirmed in hundreds of research papers. These are just some of the reasons why this method was used here. Moreover, it has not been used in the fuzzy–rough form until now. The purpose of this method is to rank alternatives in relation to the ideal and anti-ideal solution, with an alternative that is closer to the ideal and further from the anti-ideal solution being more favorable. The steps of this method are as follows:

Step 1. Forming a linguistic decision matrix. In this step, experts evaluate the alternatives based on the criteria selected. Experts assign a linguistic value to each alternative in relation to the criterion observed.

Step 2. Transforming linguistic values into fuzzy numbers.

Step 3. Determining the lower and upper limits of the rough number for each individual fuzzy number. Thus, the initial fuzzy–rough decision matrix is formed. The steps of the MARCOS method are then applied to this matrix.

Step 4. Expanding the fuzzy–rough decision matrix with an ideal and anti-ideal solution.

$$
AAI = \min_i FR_i(\overline{X_i}) \text{ for benefit criteria and } AAI = \max_i FR_i(\overline{X_i}) \text{ for cost criteria}
\tag{21}
$$

$$
AI = \max_i FR_i(\overline{X_i}) \text{ for benefit criteria and } AI = \min_i FR_i(\overline{X_i}) \text{ for cost criteria}
\tag{22}
$$

Step 5. Normalization of the expanded fuzzy–rough decision matrix. In this step, criteria and values are harmonized. This is because some criteria are of the benefit type and others are of the cost type. In the case of benefit-type criteria, the alternative must have as high a rating as possible in order to be more favorable, while in the case of cost-type criteria, the opposite is the case, i.e., it is necessary for the alternative to have as low a rating as possible. This is why two equations are used in the normalization, as follows:

$$\bar{\bar{n}}_{ij} = \left( \left[ \frac{x_{ij}^{lL}}{\max x_j^{uU}}, \frac{x_{ij}^{lU}}{\max x_j^{uL}} \right], \left[ \frac{x_{ij}^{mL}}{\max x_j^{mU}}, \frac{x_{ij}^{mU}}{\max x_j^{mL}} \right], \left[ \frac{x_{ij}^{uL}}{\max x_j^{lU}}, \frac{x_{ij}^{uU}}{\max x_j^{lL}} \right] \right) \text{ for benefit criteria} \tag{23}$$

$$\bar{\bar{n}}_{ij} = \left( \left[ \frac{\min x_j^{lL}}{x_{ij}^{uU}}, \frac{\min x_j^{lU}}{x_{ij}^{uL}} \right], \left[ \frac{\min x_j^{mL}}{x_{ij}^{mU}}, \frac{\min x_j^{mU}}{x_{ij}^{mL}} \right], \left[ \frac{\min x_{ij}^{uL}}{x_{ij}^{lU}}, \frac{\min x_j^{uU}}{x_{ij}^{lL}} \right] \right) \text{ for cost criteria} \tag{24}$$

Step 6. Weighting the normalized fuzzy–rough decision matrix. Here, the normalized fuzzy–rough decision matrix is multiplied with appropriate weights:

$$\bar{\bar{v}}_{ij} = \bar{\bar{w}}_j \cdot \bar{\bar{n}}_{ij} \tag{25}$$

Step 7. Computing the degree of utility of alternatives. Here, the degree of utility is computed in relation to the ideal and anti-ideal solution:

$$\bar{\bar{K}}_i^- = \frac{\bar{\bar{S}}_i}{\bar{\bar{S}}_{aai}} \tag{26}$$

$$\bar{\bar{K}}_i^+ = \frac{\bar{\bar{S}}_i}{\bar{\bar{S}}_{ai}} \tag{27}$$

where the values of $\bar{\bar{S}}_i$ for all alternatives, for both ideal and anti-ideal solutions, are computed as the sum of the values for all criteria.

$$\bar{\bar{S}}_i = \sum_{i=1}^{m} \bar{\bar{v}}_{ij} \tag{28}$$

Step 8. Computing the average value of the fuzzy–rough number.

$$K_i^\pm = \frac{\bar{\bar{K}}_{i1}^{\pm L} + \bar{\bar{K}}_{i1}^{\pm U} + \bar{\bar{K}}_{i2}^{\pm L} + \bar{\bar{K}}_{i2}^{\pm U} + \bar{\bar{K}}_{i3}^{\pm L} + \bar{\bar{K}}_{i3}^{\pm U}}{6} \tag{29}$$

Step 9. Determining the utility function of the alternative $f(K_i)$. When determining the utility function, a compromise in relation to the ideal and anti-ideal solution is made. This function is computed as follows:

$$f(K_i) = \frac{K_i^+ + K_i^-}{1 + \frac{1 - f(K_i^+)}{f(K_i^+)} + \frac{1 - f(K_i^-)}{f(K_i^-)}} \tag{30}$$

where

$$f(K_i^-) = \frac{K_i^+}{K_i^+ + K_i^-} \tag{31}$$

$$f(K_i^+) = \frac{K_i^-}{K_i^+ + K_i^-} \tag{32}$$

The ranking of alternatives according to the highest value.

**4. Case Study**

The Lombardija Company is engaged in the distribution of products in the territory of the Brčko District of Bosnia and Herzegovina. They are authorized distributors for several manufacturers in this area. In addition, the Lombardija Company owns three sales facilities in the territory of the Brčko District municipality of Bosnia and Herzegovina. Recently, the company opened a web shop, thus expanding the sale of products. The Lombardija Company has to perform distribution tasks daily, visiting both its own and other sales facilities and delivering goods to them. In addition, it also distributes goods directly to customers through its sales facilities and through online sales. The company is considering the possibility of distributing products using electric delivery vehicles in order to reduce its costs and the negative impact on the environment caused by the use of vehicles with internal combustion engines. In this way, the company would need to apply sustainable urban logistics, since most of the distribution is carried out in urban areas, with some occurring in rural areas.

In order to obtain information about which electric delivery vehicles would be the most suitable for a potential purchase, the company first formed a team to evaluate the current offerings on the electric delivery vehicle market. The team consisted of six experts, that is, three experts from the Lombardija Company who were engaged in product distribution and three experts from the territory of Brčko District who were traffic engineers and were familiar with the use of electric vehicles. The first expert from the company was an economics graduate with many years of experience in the logistics business who communicated with customers on a daily basis. Thus, the expert shared his knowledge based on experience with customers for the purposes of this research. The other two experts were drivers who carried out distribution. They were familiar with how distribution is carried out and had many years of experience with vehicles. However, this potential acquisition had several limitations. These limitations were related to the characteristics of electric delivery vehicles, namely: the range of these vehicles with a single charge, the charging of these vehicles, and their price. Range is one of the most limiting aspects in the application of urban logistics. Due to the scattered distribution of the sales facilities, it was necessary that the vehicle could travel at least 180 km with a single charge. The company arrived at this figure by tracking the kilometers traveled by current vehicles. Further, the territory of the Brčko District of Bosnia and Herzegovina has only one fast charger, so the batteries of the vehicles would have to be charged within the Lombardija Company facilities and by ordinary chargers. The third limitation was related to finances, namely the fact that electric delivery vehicles cost more than classic delivery vehicles. Because of this limitation, it was decided that electric vehicles with a price of more than EUR 50,000 would not be taken into consideration.

In addition to the above, when determining potential electric delivery vehicles, there was another limitation, i.e., only those vehicles with authorized service centers in Bosnia and Herzegovina were considered. In addition, combined delivery vehicles that could be transformed into cargo vehicles and passenger transport vehicles were taken into account. When choosing between vehicle variants from the same manufacturer, those with a larger cargo space were considered. Due to all these limitations imposed on the decision-making process, the following electric delivery vehicles were analyzed:

- Citroen e-Berlingo XL (V1).
- Renault Kangoo E-Tech Electric (V2).
- Peugeot e-Rifter Long (V3).
- Toyota Proace City Verso Electric L2 (V4).
- Opel Combo-e Life XL (V5).
- Mercedes EQT 200 Standard (V6).
- Opel Vivaro-e Combi M (V7).
- Citroen e-Jumpy Combi XL (V8).
- Peugeot e-Expert Combi Standard (V9).

In order to compare these electric delivery vehicles, the selected experts also chose criteria for evaluating the vehicles. All criteria are presented in Table 1.

**Table 1.** Criteria for the selection of electric delivery vehicles.

| Id | Criterion | Description | Criterion Type | Reference |
|---|---|---|---|---|
| C1 | Price | The monetary value of the vehicle | Cost | [27–30,35,36,39] |
| C2 | Acceleration | Acceleration from 0 to 100 km/h | Cost | [27–30,36,39] |
| C3 | Range | Range on a single charge | Benefit | [27–30,35,36,39] |
| C4 | Engine Power | Vehicle engine power | Benefit | [27–30,35,36,39] |
| C5 | Battery Capacity | Real battery capacity | Benefit | [27–30,35,36,39] |
| C6 | Charge Time | Charging time with regular charger | Cost | [27–30,35,36] |
| C7 | Fast Charge Time | Charging time with fast charger | Cost | [27–30,35,36] |
| C8 | Vehicle Consumption | Energy consumption per km | Cost | [27–30,36] |
| C9 | Max Payload | Maximum carrying capacity of the vehicle | Benefit | [27,35] |
| C10 | Cargo Volume | Maximum cargo space capacity | Benefit | [27,29,30,35,36,39] |

Electric delivery vehicles are more expensive than classic delivery vehicles, so the price was taken as a criterion. For the studied company, it was important to buy a vehicle that was affordable and satisfied most of the goals. Certainly, vehicles that are more expensive have more equipment included. Acceleration is important for delivery vehicles, as it is necessary to deliver goods to various places. These vehicles are battery-limited, and their range depends on their battery. If the battery is larger, the vehicle is more expensive, but at the same time it has a larger range. The power of the engine is also important because driving an empty vehicle is not the same as driving a full vehicle. In order to transport a large amount of goods, the vehicle needs to have a more powerful engine. Battery capacity is important for the range of electric delivery vehicles. The larger the battery capacity, the greater the range of the vehicle. However, this negatively affects the charging of the batteries. If the capacity is higher, the charging time is also longer, and it is desirable to reduce the charging time because it is necessary to deliver goods on time. This is why fast chargers are used, so that batteries can be charged faster. For classic vehicles, fuel consumption is important, while battery consumption is equivalent for electric vehicles. It is necessary that the consumption is lower in order to increase the range of the vehicle. In the case of delivery vehicles, the load capacity is significant, too. It is desirable that the load capacity be as high as possible. Also, the same applies to the capacity of the trunk, which should be large.

In order to use these vehicles for delivery, it is necessary to take into consideration several practical challenges and obstacles, some of which are described as follows. Electric vehicles have not yet been sufficiently accepted in practice, and there is a lack of fast charging stations for these vehicles. In addition, their price is also a limiting factor that must be considered when purchasing these vehicles. Hence, the considered company intended to use a classic electrical network and charge the batteries using slow chargers, at least according to their plans. This would be a problem because of the time it takes to charge a battery in this way. Therefore, it would be necessary to charge these vehicles only at night when they are not in use. For this reason, the company was only considering the possibility of introducing these vehicles, which did not mean that they would be introduced. This decision would be influenced by the state's decision to introduce such vehicles. Another limitation when introducing these vehicles would be their delivery time, because the Lombardija Company would become a customer, and the suppliers of the electric delivery vehicles would have to adapt to it.

In order to select an electric delivery vehicle that best met the goals set by the Lombardija Company, the following steps were taken. Step 1 was to determine the importance of the criteria and assess the selected vehicles in relation to these criteria. In order to accomplish this, linguistic ratings were used (Table 2). Step 2 was to define the membership function of the ratings to the corresponding fuzzy number. This allowed linguistic ratings

to be transformed into corresponding fuzzy numbers. Step 3 was to determine the lower and upper limits of individual fuzzy numbers using a rough set. Step 4 was to implement the selected fuzzy–rough methods for determining the weights of the criteria and the ranking of electric delivery vehicles. Step 5 was to validate the research results and conduct sensitivity analysis in order to examine the impact of certain criteria on the selection of electric delivery vehicles. At the same time, these steps represent the way in which this research was carried out.

**Table 2.** Linguistic terms and membership functions.

| For Criteria | | For Alternatives | |
|---|---|---|---|
| **Linguistic Term** | **Membership Function** | **Linguistic Terms** | **Membership Function** |
| Absolutely high (AH) | (0, 1, 2) | Absolutely low (AL) | (1, 1, 2) |
| Extremely high (EH) | (1, 2, 3) | Very low (VL) | (2, 3, 4) |
| High (H) | (2, 3, 4) | Low (L) | (3, 4, 5) |
| Medium-high (MH) | (3, 4, 5) | Medium-low (ML) | (4, 5, 6) |
| Equal (E) | (4, 5, 6) | Equal (E) | (5, 6, 7) |
| Medium-low (ML) | (5, 6, 7) | Medium-high (MH) | (6, 7, 8) |
| Low (L) | (6, 7, 8) | High (H) | (7, 8, 9) |
| Very low (VL) | (7, 8, 9) | Extremely high (EH) | (8, 9, 10) |
| Absolutely low (AL) | (8, 9, 10) | Absolutely high (AH) | (9, 10, 10) |

When evaluating alternatives according to certain criteria, it should be noted that the rating for benefit criteria should be as high as possible and the rating for cost criteria as low as possible in order for a certain alternative to be ranked higher.

## 5. Results

The first step in calculating the research results was determining the weights of the criteria. The fuzzy–rough SWARA method was used to determine the weights of the criteria. For this method, experts first determined the importance of the criteria based on linguistic ratings (Table 3). Then, these ratings were transformed into fuzzy numbers using the membership function.

**Table 3.** Initial linguistic fuzzy decision matrix.

| | DM1 | DM2 | DM3 | DM4 | DM5 | DM6 |
|---|---|---|---|---|---|---|
| C1 | EH (1, 2, 3) | H (2, 3, 4) | EH (1, 2, 3) | EH (1, 2, 3) | AH (0, 1, 2) | AH (0, 1, 2) |
| C2 | EH (1, 2, 3) | MH (3, 4, 5) | EH (1, 2, 3) | H (2, 3, 4) | E (4, 5, 6) | MH (3, 4, 5) |
| C3 | AH (0, 1, 2) | AH (0, 1, 2) | AH (0, 1, 2) | AH (0, 1, 2) | MH (3, 4, 5) | H (2, 3, 4) |
| C4 | EH (1, 2, 3) | MH (3, 4, 5) | EH (1, 2, 3) | AH (0, 1, 2) | MH (3, 4, 5) | H (2, 3, 4) |
| C5 | EH (1, 2, 3) | E (4, 5, 6) | AH (0, 1, 2) | AH (0, 1, 2) | EH (1, 2, 3) | H (2, 3, 4) |
| C6 | EH (1, 2, 3) | EH (1, 2, 3) | H (2, 3, 4) | EH (1, 2, 3) | EH (1, 2, 3) | MH (3, 4, 5) |
| C7 | EH (1, 2, 3) | MH (3, 4, 5) | EH (1, 2, 3) | EH (1, 2, 3) | H (2, 3, 4) | EH (1, 2, 3) |
| C8 | EH (1, 2, 3) | EH (1, 2, 3) | H (2, 3, 4) | H (2, 3, 4) | H (2, 3, 4) | E (4, 5, 6) |
| C9 | EH (1, 2, 3) | EH (1, 2, 3) | EH (1, 2, 3) | EH (1, 2, 3) | H (3, 4, 5) | MH (3, 4, 5) |
| C10 | AH (0, 1, 2) | AH (0, 1, 2) | AH (0, 1, 2) | EH (1, 2, 3) | H (3, 4, 5) | H (2, 3, 4) |

After the linguistic values were transformed into fuzzy numbers, fuzzy–rough numbers were calculated. The example of C1 will explain how these numbers were formed.

For fuzzy number "*l*", rough limits were formed as follows:

$\underline{Lim}DM1 = \frac{1+1+1+0+0}{5} = 0.60$, $\underline{Lim}DM2 = \frac{1+2+1+1+0+0}{6} = 0.83$, $\underline{Lim}DM3 = \frac{1+1+1+0+0}{5} = 0.60$, $\underline{Lim}DM4 = \frac{1+1+1+0+0}{5} = 0.60$, $\underline{Lim}DM5 = \frac{0+0}{2} = 0.00$, $\underline{Lim}DM6 = \frac{0+0}{2} = 0.00$.

$\overline{Lim}DM1 = \frac{1+2+1+1}{4} = 1.25$, $\overline{Lim}DM2 = \frac{2}{1} = 2$, $\overline{Lim}DM3 = \frac{1+2+1+1}{4} = 1.25$, $\overline{Lim}DM4 = \frac{1+2+1+1}{4} = 1.25$, $\overline{Lim}DM5 = \frac{1+2+1+1+0+0}{6} = 0.83$, $\overline{Lim}DM6 = \frac{1+2+1+1+0+0}{6} = 0.83$.

For fuzzy number "*m*", rough limits were formed as follows:

$\underline{Lim}DM1 = \frac{2+2+2+1+1}{5} = 1.60$, $\underline{Lim}DM2 = \frac{2+3+2+2+1+1}{6} = 1.83$, $\underline{Lim}DM3 = \frac{2+2+2+1+1}{5} = 1.60$, $\underline{Lim}DM4 = \frac{2+2+2+1+1}{5} = 1.60$, $\underline{Lim}DM5 = \frac{1+1}{2} = 1.00$, $\underline{Lim}DM6 = \frac{1+1}{2} = 1.00$.

$\overline{Lim}DM1 = \frac{2+3+2+2}{4} = 2.25$, $\overline{Lim}DM2 = \frac{3}{1} = 3$, $\overline{Lim}DM3 = \frac{2+3+2+2}{4} = 2.25$, $\overline{Lim}DM4 = \frac{2+3+2+2}{4} = 2.25$, $\overline{Lim}DM5 = \frac{2+3+2+2+1+1}{6} = 1.83$, $\overline{Lim}DM6 = \frac{2+3+2+2+1+1}{6} = 1.83$.

For fuzzy number "*u*", rough limits were formed as follows:

$\underline{Lim}DM1 = \frac{3+3+3+2+2}{5} = 2.60$, $\underline{Lim}DM2 = \frac{3+4+3+3+2+2}{6} = 2.83$, $\underline{Lim}DM3 = \frac{3+3+3+2+2}{5} = 2.60$, $\underline{Lim}DM4 = \frac{3+3+3+2+2}{5} = 2.60$, $\underline{Lim}DM5 = \frac{2+2}{2} = 2.00$, $\underline{Lim}DM6 = \frac{2+2}{2} = 2.00$.

$\overline{Lim}DM1 = \frac{3+4+3+3}{4} = 3.25$, $\overline{Lim}DM2 = \frac{4}{1} = 4$, $\overline{Lim}DM3 = \frac{3+4+3+3}{4} = 3.25$, $\overline{Lim}DM4 = \frac{3+4+3+3}{4} = 3.25$, $\overline{Lim}DM5 = \frac{3+4+3+3+2+2}{6} = 2.83$, $\overline{Lim}DM6 = \frac{3+4+3+3+2+2}{6} = 2.83$.

The final values for the fuzzy–rough decision matrix were obtained by computing the average values for all decision makers (Table 4).

**Table 4.** Initial fuzzy–rough decision matrix for the SWARA method.

| | *Xj* | | *Xj* |
|---|---|---|---|
| C1 | [(0.44, 1.21), (1.44, 2.21), (2.44, 3.24)] | C3 | [(0.21, 1.09), (1.21, 2.09), (2.21, 3.47)] |
| C2 | [(1.61, 2.61), (2.61, 3.61), (3.61, 5.05)] | C1 | [(0.44, 1.21), (1.44, 2.21), (2.44, 3.24)] |
| C3 | [(0.21, 1.09), (1.21, 2.09), (2.21, 3.47)] | C10 | [(0.31, 1.31), (1.31, 2.31), (2.31, 3.75)] |
| C4 | [(0.95, 1.95), (1.95, 2.95), (2.95, 4.39)] | C5 | [(0.52, 1.52), (1.52, 2.52), (2.52, 4.28)] |
| C5 | [(0.52, 1.52), (1.52, 2.52), (2.52, 4.28)] | C6 | [(1.12, 1.78), (2.12, 2.78), (3.12, 3.92)] |
| C6 | [(1.12, 1.78), (2.12, 2.78), (3.12, 3.92)] | C7 | [(1.12, 1.78), (2.12, 2.78), (3.12, 3.92)] |
| C7 | [(1.12, 1.78), (2.12, 2.78), (3.12, 3.92)] | C4 | [(0.95, 1.95), (1.95, 2.95), (2.95, 4.39)] |
| C8 | [(1.47, 2.42), (2.47, 3.42), (3.47, 4.58)] | C9 | [(1.22, 2.00), (2.22, 3.00), (3.22, 4.11)] |
| C9 | [(1.22, 2.00), (2.22, 3.00), (3.22, 4.11)] | C8 | [(1.47, 2.42), (2.47, 3.42), (3.47, 4.58)] |
| C10 | [(0.31, 1.31), (1.31, 2.31), (2.31, 3.75)] | C2 | [(1.61, 2.61), (2.61, 3.61), (3.61, 5.05)] |

When this initial decision matrix was formed (Table 4), it was necessary to order the criteria by their values, with the best criterion being that with the lowest value in this decision matrix. In this case, it was criterion C3. This was followed by criterion C1 and C10, respectively, while criterion C2 had the highest values. After this, the data normalization of the initial fuzzy–rough decision matrix was performed. Criterion C3 obtained a value of one at all limits, while the other values were divided by the maximum value for all criteria, which were the values of criterion C2 [(1.61, 2.61), (2.61, 3.61), (3.61, 5.05)]. For the example of criterion C1, its values were formed as follows: $N_j = \left[ \left( \frac{0.44}{5.05}, \frac{1.21}{3.61} \right), \left( \frac{1.44}{3.61}, \frac{2.21}{2.61} \right), \left( \frac{2.44}{2.61}, \frac{3.24}{1.61} \right) \right] = [(0.08, 0.33), (0.40, 0.84), (0.93, 2.00)]$. The values for other criteria were calculated in the same way. Thus, a normalized decision matrix $FRN$ ($N_j$) was formed.

Then, the value of criterion C3 was copied, while the value of 1 was added to the other criteria. In this way, the matrix $FRN(\Im_j)$ was formed. For criterion C1, it was formed in the following way: $[(0.08, 0.33), (0.40, 0.84), (0.93, 2.00)] + [(1, 1), (1, 1), (1, 1)] = [(1.08, 1.33), (1.40, 1.84), (1.93, 3.00)]$. The values for the other criteria were formed in the same way. This matrix was created by copying the value of the first criterion once again, in this case criterion C3, while the values of the other criteria were formed by dividing the value of $FRN(\Im_j)$ for that criterion by the value of $FRN(\Re_j)$ for the previous criterion (Table 5). For the example of criterion C1, this process was carried out as follows: $FRN(C_1) = \left[ \left( \frac{1.00}{3.00}, \frac{1.00}{1.93} \right), \left( \frac{1.00}{1.84}, \frac{1.00}{1.40} \right), \left( \frac{1.00}{1.33}, \frac{1.00}{1.08} \right) \right] = [(0.33, 0.52), (0.54, 0.72), (0.75, 0.92)]$.

Then, the addition of all the values of this newly formed matrix $FRN(\Re_j)$ was performed. After this, the final weight values for the criteria were determined by forming a matrix $FRN(W_j)$. This matrix was formed by dividing the individual values of the criteria in the matrix $FRN(\Re_j)$ by the total value of this matrix for all criteria (Table 5). For the example of C1, this was carried out as follows: $FRN(W_{C1}) = \left[ \left( \frac{0.33}{6.03}, \frac{0.52}{3.37} \right), \left( \frac{0.54}{3.20}, \frac{0.72}{2.11} \right), \left( \frac{0.75}{2.05}, \frac{0.92}{1.47} \right) \right] =$ $[(0.055, 0.153), (0.169, 0.339), (0.366, 0.626)]$. The weights for all criteria were calculated in the same way. Thus, the weights of the criteria were formed. Criterion C3 (range) had the greatest weight, followed by C1 (price) and C10 (cargo volume), while criterion C2 (acceleration) had the lowest weight.

**Table 5.** Final values of criteria weights.

| | $FRN(\Re_j)$ | $FRN(W_j)$ |
|---|---|---|
| C3 | [(1.000, 1.000), (1.000, 1.000), (1.000, 1.000)] | [(0.166, 0.297), (0.313, 0.474), (0.489, 0.680)] |
| C1 | [(0.332, 0.517), (0.542, 0.715), (0.749, 0.920)] | [(0.055, 0.153), (0.169, 0.339), (0.366, 0.626)] |
| C10 | [(0.100, 0.274), (0.288, 0.525), (0.550, 0.867)] | [(0.017, 0.081), (0.090, 0.249), (0.269, 0.590)] |
| C5 | [(0.027, 0.140), (0.146, 0.369), (0.387, 0.786)] | [(0.005, 0.041), (0.046, 0.175), (0.189, 0.535)] |
| C6 | [(0.008, 0.064), (0.071, 0.233), (0.259, 0.644)] | [(0.001, 0.019), (0.022, 0.110), (0.127, 0.438)] |
| C7 | [(0.002, 0.029), (0.034, 0.147), (0.173, 0.527)] | [(0.000, 0.009), (0.011, 0.070), (0.085, 0.359)] |
| C4 | [(0.001, 0.014), (0.016, 0.095), (0.113, 0.444)] | [(0.000, 0.004), (0.005, 0.045), (0.055, 0.302)] |
| C9 | [(0.000, 0.006), (0.007, 0.059), (0.072, 0.358)] | [(0.000, 0.002), (0.002, 0.028), (0.035, 0.243)] |
| C8 | [(0.000, 0.003), (0.003, 0.035), (0.043, 0.277)] | [(0.000, 0.001), (0.001, 0.017), (0.021, 0.188)] |
| C2 | [(0.000, 0.001), (0.001, 0.020), (0.025, 0.210)] | [(0.000, 0.000), (0.000, 0.010), (0.012, 0.143)] |
| | [(1.471, 2.047), (2.109, 3.198), (3.373, 6.034)] | |

After the weights of the criteria used in this analysis were calculated, the studied electric delivery vehicles were evaluated. The first step was to evaluate the selected criteria in relation to the observed alternatives. Here, experts evaluated these criteria using linguistic values (Table 6).

**Table 6.** Initial linguistic decision matrix for alternatives.

| DM1 | C1 | C2 | C3 | C4 | C5 | C6 | C7 | C8 | C9 | C10 | DM2 | C1 | C2 | C3 | C4 | C5 | C6 | C7 | C8 | C9 | C10 |
|---|---|---|---|---|---|---|---|---|---|---|---|---|---|---|---|---|---|---|---|---|---|
| V1 | L | ML | ML | M | M | EH | ML | M | M | L | V1 | L | M | M | MH | M | H | L | M | M | M |
| V2 | ML | MH | M | ML | ML | MH | H | ML | L | M | V2 | ML | MH | MH | MH | M | M | MH | ML | ML | MH |
| V3 | ML | ML | ML | M | M | EH | ML | M | M | L | V3 | ML | M | M | MH | M | H | L | M | M | M |
| V4 | ML | VL | ML | M | M | EH | ML | M | ML | MH | V4 | ML | ML | M | MH | M | H | L | M | ML | MH |
| V5 | M | ML | ML | M | M | EH | ML | MH | M | MH | V5 | M | H | M | MH | M | H | L | M | M | MH |
| V6 | H | MH | M | ML | ML | L | H | L | ML | VL | V6 | H | MH | MH | M | M | VL | MH | ML | ML | ML |
| V7 | H | M | VL | M | M | EH | ML | H | EH | EH | V7 | H | MH | ML | MH | M | H | L | MH | EH | H |
| V8 | H | M | VL | M | M | EH | ML | EH | EH | AH | V8 | EH | MH | ML | MH | M | H | L | MH | H | EH |
| V9 | H | H | VL | M | M | EH | ML | H | EH | EH | V9 | H | H | ML | MH | M | H | L | MH | H | H |

| DM3 | C1 | C2 | C3 | C4 | C5 | C6 | C7 | C8 | C9 | C10 | DM4 | C1 | C2 | C3 | C4 | C5 | C6 | C7 | C8 | C9 | C10 |
|---|---|---|---|---|---|---|---|---|---|---|---|---|---|---|---|---|---|---|---|---|---|
| V1 | VL | L | MH | EH | H | EH | L | M | H | ML | V1 | M | MH | EH | EH | H | H | H | VL | EH | H |
| V2 | L | ML | H | H | MH | H | ML | ML | MH | M | V2 | MH | H | AH | H | EH | M | EH | L | MH | EH |
| V3 | L | L | MH | EH | H | EH | L | M | H | ML | V3 | MH | MH | EH | EH | EH | H | H | VL | H | H |
| V4 | L | L | MH | EH | H | EH | L | M | MH | MH | V4 | M | M | EH | EH | EH | H | H | VL | MH | EH |
| V5 | ML | L | M | EH | H | EH | L | M | H | MH | V5 | MH | MH | H | H | EH | H | H | VL | EH | H |
| V6 | M | ML | EH | H | MH | M | ML | L | MH | M | V6 | ML | H | AH | H | EH | L | EH | VL | MH | MH |
| V7 | M | L | ML | EH | H | EH | L | MH | EH | H | V7 | MH | M | H | H | H | H | H | L | EH | H |
| V8 | M | L | ML | EH | H | EH | L | MH | EH | EH | V8 | H | MH | H | H | H | H | H | VL | EH | EH |
| V9 | M | M | ML | EH | H | EH | L | MH | EH | H | V9 | EH | H | MH | EH | H | H | H | L | EH | H |

| DM5 | C1 | C2 | C3 | C4 | C5 | C6 | C7 | C8 | C9 | C10 | DM6 | C1 | C2 | C3 | C4 | C5 | C6 | C7 | C8 | C9 | C10 |
|---|---|---|---|---|---|---|---|---|---|---|---|---|---|---|---|---|---|---|---|---|---|
| V1 | VL | L | M | MH | MH | M | L | ML | MH | MH | V1 | AL | ML | M | M | M | ML | VL | L | M | M |
| V2 | VL | ML | MH | MH | ML | L | ML | L | M | EH | V2 | VL | MH | MH | ML | ML | L | ML | L | M | MH |
| V3 | VL | L | H | MH | MH | M | L | ML | MH | MH | V3 | L | ML | M | M | M | ML | VL | L | M | M |
| V4 | L | VL | M | MH | MH | M | L | ML | M | H | V4 | ML | L | M | M | M | ML | VL | L | M | MH |
| V5 | ML | L | M | MH | MH | M | L | ML | MH | H | V5 | M | ML | M | M | M | ML | VL | L | M | MH |
| V6 | M | M | MH | M | ML | AL | ML | VL | M | M | V6 | M | MH | H | ML | ML | AL | ML | AL | M | ML |
| V7 | ML | ML | M | M | M | M | L | L | H | EH | V7 | ML | M | ML | ML | M | ML | VL | L | EH | MH |
| V8 | ML | ML | M | M | M | M | L | L | H | AH | V8 | M | M | M | ML | M | ML | VL | L | H | H |
| V9 | ML | M | M | M | M | M | L | L | H | EH | V9 | M | H | ML | M | M | ML | VL | L | H | MH |

After this, the same steps as with the fuzzy–rough SWARA method were conducted, that is, the transformation of linguistic values into fuzzy numbers and the calculation of

the lower and upper limits of the fuzzy–rough numbers. The only difference lay in the membership function that was used for the alternatives (Table 2). Applying the same rules as with the fuzzy–rough SWARA method, an initial fuzzy–rough decision matrix was created. The matrix was expanded, and an ideal and anti-ideal solution were determined (Table 7). In this case, the ideal solution was the minimum value of the alternatives if referring to cost criteria, or the maximum value of the alternatives if referring to benefit criteria. The opposite was the case for determining the anti-ideal solution.

**Table 7.** Expanded initial fuzzy–rough decision matrix for alternatives.

| | C1 | C2 | ... | C10 |
|---|---|---|---|---|
| AAI | [(5.09, 6.91) (6.91, 7.91) (7.91, 8.91)] | [(5.89, 6.78) (6.89, 7.78) (7.89, 8.78)] | ... | [(3.50, 5.10) (5.10, 6.10) (6.10, 7.10)] |
| V1 | [(1.90, 3.50) (3.50, 4.47) (4.47, 7.77)] | [(3.50, 4.89) (4.89, 5.89) (6.09, 7.91)] | ... | [(4.10, 5.90) (5.90, 6.90) (6.90, 8.75)] |
| V2 | [(2.64, 4.43) (4.43, 5.43) (5.43, 6.43)] | [(4.85, 6.13) (6.13, 7.13) (7.13, 8.13)] | ... | [(5.61, 7.11) (7.11, 8.11) (8.11, 9.11)] |
| V3 | [(2.90, 4.50) (4.50, 5.50) (5.50, 6.50)] | [(3.50, 4.89) (4.89, 5.89) (5.89, 6.89)] | ... | [(4.10, 5.90) (5.90, 6.90) (6.90, 7.90)] |
| V4 | [(3.44, 4.24) (4.44, 5.24) (5.44, 6.24)] | [(2.50, 3.89) (3.89, 4.89) (4.89, 5.89)] | ... | [(6.12, 6.92) (7.12, 7.92) (8.12, 8.92)] |
| V5 | [(4.44, 5.24) (5.44, 6.24) (6.44, 7.24)] | [(3.58, 5.50) (5.50, 6.50) (6.50, 7.50)] | ... | [(6.11, 6.55) (7.11, 7.55) (8.11, 8.55)] |
| V6 | [(4.88, 6.15) (6.15, 7.15) (7.15, 8.15)] | [(5.06, 6.24) (6.24, 7.24) (7.24, 8.24)] | ... | [(3.50, 5.10) (5.10, 6.10) (6.10, 7.10)] |
| V7 | [(4.68, 6.32) (6.32, 7.32) (7.32, 8.32)] | [(4.06, 5.24) (5.24, 6.24) (6.24, 7.24)] | ... | [(6.77, 7.56) (7.77, 8.56) (8.77, 9.56)] |
| V8 | [(5.09, 6.91) (6.91, 7.91) (7.91, 8.91)] | [(4.11, 5.51) (5.51, 6.51) (6.51, 7.51)] | ... | [(7.77, 8.56) (8.77, 9.56) (9.69, 9.97)] |
| V9 | [(5.09, 6.91) (6.91, 7.91) (7.91, 8.91)] | [(5.89, 6.78) (6.89, 7.78) (7.89, 8.78)] | ... | [(6.77, 7.56) (7.77, 8.56) (8.77, 9.56)] |
| AI | [(1.90, 3.50) (3.50, 4.47) (4.47, 6.24)] | [(2.50, 3.89) (3.89, 4.89) (4.89, 5.89)] | ... | [(7.77, 8.56) (8.77, 9.56) (9.69, 9.97)] |

The next step in the implementation of the fuzzy–rough MARCOS method was the normalization of this expanded initial fuzzy–rough decision matrix. The normalization procedure differed depending on the type of criteria, i.e., whether they were cost or benefit criteria (Equations (23) and (24)). For the example of vehicle 1, the normalization of criterion 1 was conducted as follows: First, the minimum value of the criterion was calculated, since this criterion was of the cost type. This produced the value [(1.9, 3.5) (3.5, 4.5) (4.5, 6.2)]. The value of V1 was then divided by this value, and the normalization was calculated as $n_{ij} = \left[ \left( \frac{1.9}{7.8}, \frac{3.5}{4.5} \right), \left( \frac{3.5}{4.5}, \frac{4.5}{3.5} \right), \left( \frac{4.5}{3.5}, \frac{6.2}{1.9} \right) \right] = [(0.24, 0.78), (0.78, 1.28), (1.28, 3.28)]$. Normalization was conducted in the same way for other criteria and alternatives. After this, the weighting of the normalized decision matrix was carried out. This was performed by multiplying the weights by the normalized values of the fuzzy–rough decision matrix. For the example of vehicle 1, criterion 1 was calculated as follows:

$$[(0.24, 0.78), (0.78, 1.28), (1.28, 3.28)] \times [(0.06, 0.15), (0.17, 0.34), (0.37, 0.63)]$$
$$= [(0.01, 0.12), (0.13, 0.43), (0.47, 2.05)]$$

The weighted decision matrix for other alternatives and criteria was calculated in the same way. Next came the determination of the degree of utility (Table 8). Before this was carried out, the sums of the alternatives were calculated for all criteria, including ideal and anti-ideal solutions. The degree of utility of the anti-ideal solution was calculated in such a way that the value of the sum for the specified alternative was divided by the anti-ideal solution (Equation (26)). The degree of utility of the ideal solution was calculated in the same way but only divided by the ideal solution (Equation (27)). For the example of vehicle 1, the process was as follows: $K_1^- = \left[ \left( \frac{0.11}{4.94}, \frac{0.44}{1.26} \right), \left( \frac{0.47}{1.15}, \frac{1.52}{0.36} \right), \left( \frac{1.65}{0.33}, \frac{7.02}{0.081} \right) \right] = [(0.022, 0.348), (0.414, 4.261), (5.062, 86.962)]$, $K_1^+ = \left[ \left( \frac{0.11}{8.70}, \frac{0.44}{1.95} \right), \left( \frac{0.47}{1.78}, \frac{1.52}{0.57} \right), \left( \frac{1.65}{0.52}, \frac{7.02}{0.14} \right) \right] [(0.012, 0.224), (0.267, 2.678), (3.189, 50.773)]$. After this, utility functions were formed (Equations (31) and (32)).

**Table 8.** Degree and function of utility for the MARCOS method.

| | $\overline{\overline{K}}_i^-$ | $\overline{\overline{K}}_i^+$ |
|---|---|---|
| V1 | [(0.022, 0.348), (0.414, 4.261), (5.062, 86.962)] | [(0.012, 0.224), (0.267, 2.678), (3.189, 50.773)] |
| V2 | [(0.026, 0.365), (0.433, 4.216), (4.999, 78.165)] | [(0.015, 0.235), (0.279, 2.649), (3.15, 45.637)] |
| V3 | [(0.023, 0.344), (0.409, 4.096), (4.863, 79.34)] | [(0.013, 0.222), (0.264, 2.574), (3.064, 46.323)] |
| V4 | [(0.023, 0.339), (0.409, 4.11), (4.957, 77.885)] | [(0.013, 0.218), (0.264, 2.583), (3.123, 45.474)] |
| V5 | [(0.023, 0.306), (0.375, 3.78), (4.612, 72.509)] | [(0.013, 0.197), (0.242, 2.376), (2.905, 42.335)] |
| V6 | [(0.026, 0.349), (0.414, 4.041), (4.793, 81.267)] | [(0.015, 0.225), (0.267, 2.54), (3.019, 47.448)] |
| V7 | [(0.018, 0.293), (0.352, 3.654), (4.401, 71.216)] | [(0.01, 0.189), (0.227, 2.296), (2.772, 41.58)] |
| V8 | [(0.018, 0.3), (0.36, 3.702), (4.447, 71.248)] | [(0.01, 0.193), (0.233, 2.326), (2.801, 41.599)] |
| V9 | [(0.018, 0.279), (0.335, 3.527), (4.253, 69.12)] | [(0.01, 0.179), (0.216, 2.217), (2.679, 40.356)] |
| | $\overline{\overline{f(K_i^-)}}$ | $\overline{\overline{f(K_i^+)}}$ |
| V1 | [(0, 0.009), (0.01, 0.104), (0.124, 1.975)] | [(0.001, 0.014), (0.016, 0.166), (0.197, 3.382)] |
| V2 | [(0.001, 0.009), (0.011, 0.103), (0.122, 1.775)] | [(0.001, 0.014), (0.017, 0.164), (0.194, 3.04)] |
| V3 | [(0.001, 0.009), (0.01, 0.1), (0.119, 1.802)] | [(0.001, 0.013), (0.016, 0.159), (0.189, 3.086)] |
| V4 | [(0.001, 0.008), (0.01, 0.1), (0.121, 1.769)] | [(0.001, 0.013), (0.016, 0.16), (0.193, 3.029)] |
| V5 | [(0, 0.008), (0.009, 0.092), (0.113, 1.646)] | [(0.001, 0.012), (0.015, 0.147), (0.179, 2.82)] |
| V6 | [(0.001, 0.009), (0.01, 0.099), (0.117, 1.845)] | [(0.001, 0.014), (0.016, 0.157), (0.186, 3.161)] |
| V7 | [(0, 0.007), (0.009, 0.089), (0.108, 1.617)] | [(0.001, 0.011), (0.014, 0.142), (0.171, 2.77)] |
| V8 | [(0, 0.008), (0.009, 0.09), (0.109, 1.618)] | [(0.001, 0.012), (0.014, 0.144), (0.173, 2.771)] |
| V9 | [(0, 0.007), (0.008, 0.086), (0.104, 1.569)] | [(0.001, 0.011), (0.013, 0.137), (0.165, 2.688)] |

When all the necessary parameters were calculated, the values of the fuzzy–rough MARCOS method were computed. Before this was carried out, the fuzzy–rough numbers were transformed into crisp numbers (Equation (29)). This was achieved with a simple arithmetic mean, that is, the average values for individual elements of the fuzzy–rough number were computed. Finally, the value of the fuzzy–rough MARCOS method was calculated (Table 9). Based on the results of this method, the best-ranked electric delivery vehicle according to expert ratings was V1 (Citroen e-Berlingo XL), followed by V6 (Mercedes EQT 200 Standard), while the worst-rated vehicle was V9 (Peugeot e-Expert Combi Standard).

**Table 9.** Results of the fuzzy–rough MARCOS method.

| | $K_i^-$ | $K_i^+$ | $f(K_i^-)$ | $f(K_i^+)$ | $f(K_i)$ | Rank |
|---|---|---|---|---|---|---|
| V1 | 16.178 | 9.524 | 0.370 | 0.636 | 7.854 | 1 |
| V2 | 14.701 | 8.661 | 0.337 | 0.579 | 6.320 | 4 |
| V3 | 14.846 | 8.743 | 0.340 | 0.584 | 6.457 | 3 |
| V4 | 14.621 | 8.612 | 0.335 | 0.575 | 6.239 | 5 |
| V5 | 13.601 | 8.011 | 0.312 | 0.535 | 5.298 | 6 |
| V6 | 15.148 | 8.919 | 0.347 | 0.596 | 6.758 | 2 |
| V7 | 13.322 | 7.846 | 0.305 | 0.524 | 5.056 | 8 |
| V8 | 13.346 | 7.860 | 0.306 | 0.525 | 5.078 | 7 |
| V9 | 12.922 | 7.610 | 0.296 | 0.508 | 4.722 | 9 |

In order to confirm the quality of their models, more and more authors have included in their research a comparative analysis of their results with results obtained using other methods, such as in [53–56]. In order to test the results of this research, other MCDM methods in the fuzzy–rough form were applied, namely SAW (Simple Additive Weighting), ARAS (Additive Ratio Assessment), CRADIS, MABAC, and WPM (Weighted Product Method). Their vehicle rankings were compared with the results of the fuzzy–rough MARCOS method. Thus, the results were validated. As can be seen, the fuzzy–rough MABAC method produced the most deviations in the rankings (Figure 2). This was because this method applied different normalization and weighting approaches to the normalized decision matrix. Hence, the ranking of this method was also different from that of the other methods. However, it cannot be said that the results of the MARCOS method were equal

to those of the other methods. They differed, but the difference was not as significant as that of the fuzzy–rough MABAC method. Thus, the greatest difference in the rankings was found for vehicles V2 and V6, which had four different rankings according to the six methods. For the other vehicles, the ranking was more uniform. Based on this, it could be concluded that the results of the fuzzy–rough MARCOS method were acceptable, since the results were not the same for any fuzzy–rough method that was applied. This deviation was caused by the criteria used, so a sensitivity analysis was conducted to investigate how much each criterion affected the change in the ranking of the alternatives.

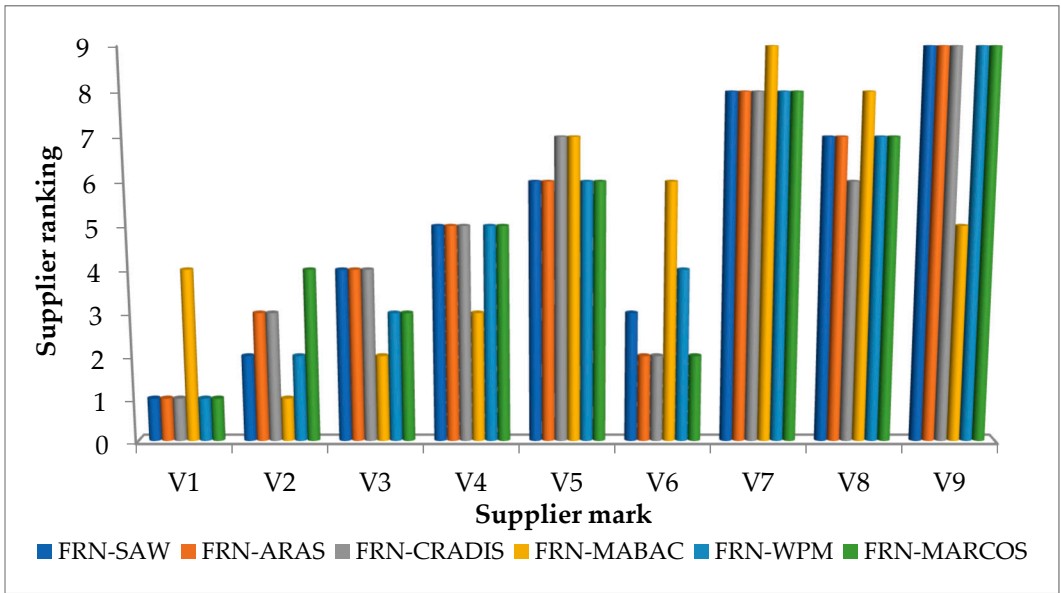

**Figure 2.** Validation of the results.

As well as a comparison with other MCDM methods, the sensitivity analysis of the model played a significant role in the validation of the results obtained [57–60]. Sensitivity analysis aims to examine how much the application of the weight of one criterion affects the final ranking of the studied alternatives [61], in this case vehicles. In this analysis, the value of one criterion was reduced by 5%, 15%, 25%, etc., up to 95%. Accordingly, 10 scenarios were created for one criterion. Since there were 10 criteria, 100 scenarios were created. With a decrease in the weight of one criterion, it is necessary to increase the weights of the other criteria proportionally. This is performed by applying the following equation:

$$W_{n\beta} = (1 - W_{n\alpha}) \frac{W_\beta}{(1 - W_n)} \tag{33}$$

The results of this analysis showed that only in the case of V9 was there no change in the ranking in all scenarios, that is, this vehicle took last place (Figure 2). Other vehicles changed their ranking when the criterion weights changed. Thus, V1 showed sensitivity to the change in the weight of criterion C1 (price). In these scenarios, V6 took first place in the ranking. However, V6 showed the greatest sensitivity to the change in the weight of criterion C6 (charge time). This was because this vehicle had the shortest charging time compared to the other vehicles, so with the change in the weight of this criterion, V6 took the penultimate place. The situation was similar for the other vehicles (Figure 3). The sensitivity analysis showed how to make certain electric delivery vehicles as favorable as possible. Thus, V6 had a higher price compared to V1 according to the experts' estimates, so it would be necessary to reduce the price of this vehicle in order for it to be ranked higher. Therefore, certain manufacturers could use the results from the sensitivity analysis in order to improve their sales.

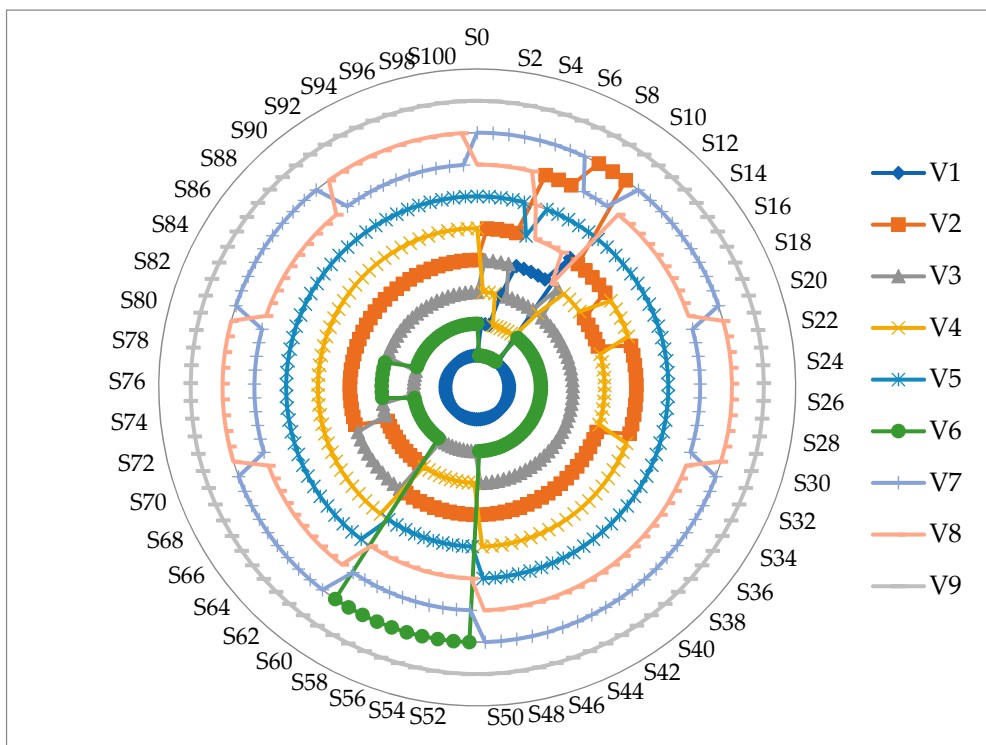

**Figure 3.** Results of sensitivity analysis.

## 6. Discussion

The increasing number of people in urban areas has made it difficult to carry out distribution [62]. In addition, there has been an expansion of online shopping, which has increased the amount of goods to be delivered. For this reason, new vehicles and new ways of delivering goods are being introduced in urban areas. Electric delivery vehicles are increasingly being used. These vehicles are also used due to the high air pollution in urban areas caused by the use of a large number of transportation vehicles [11]. Hence, the application of sustainable urban logistics, which aims to reduce the impact on the environment, especially in last-mile delivery, is increasingly advocated for in practice. Last-mile delivery increases the cost of delivery and makes it more complex [15,19,63]. Based on this, this study designed a selection process for an electric delivery vehicle, considering classic electric vehicles that are multifunctional and can transport people and goods. These vehicles can be used for different purposes: if required, it is possible to transport employees, and if not, then the maximum luggage space can be used for the delivery of goods. In this way, electric delivery vehicles can change their purpose, and it is not necessary to use another vehicle when people must be transported.

The research discussed the example of the procurement of an electric delivery vehicle to meet the needs of the Lombardija Company, Brčko. This company is a distributor of well-known international and domestic brands in the territory of the Brčko District of Bosnia and Herzegovina, and it owns its sales facilities and provides online sales. Thus, this company delivers goods to its sales facilities, to the sales facilities of other companies, and directly to customers. Thus, goods are distributed daily in the territory of the Brčko District of Bosnia and Herzegovina. In order to reduce costs by introducing electric delivery vehicles [18], the company considered the introduction of such vehicles for distribution. For this purpose, experts were selected and initial restrictions were set, that is, the vehicles could cost no more than EUR 50,000, and they had to be able to travel at least 180 km with one charge. In addition, these vehicles were required to have an authorized service center in the territory of Bosnia and Herzegovina. Thus, nine different electric vehicles that could be used as delivery vehicles were taken into consideration.

These limitations are in fact also the general limitations that apply to electric vehicles. Electric vehicles are more expensive than classic vehicles [64] and can travel shorter distances with one charge than classic vehicles with one tank of fuel [65]. In addition, it takes much longer to charge electric vehicle batteries than it does to refuel. It is the charging of the battery that is crucial, which is why the limit of 180 km on a single charge was considered. There is one fast charging station for electric vehicles in the Brčko District of Bosnia and Herzegovina, and the Lombardija Company would have to charge their vehicles using classic chargers, which take much longer to charge batteries. The problem of the lack of charging stations for electric vehicles exists not only in Bosnia and Herzegovina, but also worldwide. Therefore, when using these vehicles, route optimization must be carried out in order to overcome these limitations in the implementation of sustainable urban logistics [26].

In order to select electric delivery vehicles, experts chose the criteria by which they assessed the selected vehicles. Then, they evaluated the importance of these criteria. The fuzzy–rough SWARA method was used to determine the importance of the criteria. The results of this approach showed that the most important criteria for electric delivery vehicles for the Lombardija Company were the range, price, and cargo volume of these vehicles. In contrast, in the research by Štilić et al. [29], where electric taxi vehicles were investigated, the charging time, fast charging time, and range were found to be the most important criteria, while the weight of the vehicle and the power of the vehicle were the most important criteria according to the research by Puška et al. [36]. Based on this, it can be concluded that the individual importance of criteria is determined by the purpose of electric vehicles. It was important that vehicles could cover a larger distance with one charge, that the price of the vehicles was not too high, and that the delivery of goods could be carried out. The results indicated that the carrying capacity of the vehicles, the consumption of these vehicles, and their acceleration were not crucial to the experts. This was because the company mainly transported food and drinks to sales facilities, and vehicles with a large carrying capacity were not necessary. In addition, the consumption of these vehicles did not play a significant role, because they consume less than classic vehicles, especially for urban logistics [16], and are increasingly employed worldwide.

These criterion weights had a great influence on the selection of a potential electric delivery vehicle to meet the needs of Lombardija Company, Brčko. Thus, by applying the fuzzy–rough MARCOS method, the electric vehicle Citroen e-Berlingo XL was found to have the best characteristics. In addition to this vehicle, the next choice was the Mercedes EQT 200 Standard vehicle, which had the shortest charging time using the classic charger that could be implemented by the Lombardija Company. These results were confirmed by validation and sensitivity analysis. Through the validation of the results in this research, in addition to comparing the results of different methods, the methods were also compared. The comparison showed that different methods provided different results precisely because of the steps they applied. The greatest difference was shown by the MABAC method, and this was because this method used a different data normalization approach, and the ranking was based on the average value of the alternatives. In addition, the weighting process was different compared to other methods.

The methodology used in this research was novel and as such had to be compared with other approaches. The basis of decision making is that the decision maker has all the information regarding the decision, and so it is necessary to apply different methods and procedures. This study provided a way in which the fuzzy–rough approach can be used in decision making. In future research, this approach must be compared with other approaches in order to establish its advantages and disadvantages. Some of the advantages of this method are the combination of fuzzy and rough approaches, exploiting the advantages of both; the development of a new approach using the MARCOS method; and the inclusion of uncertainty in decision making. However, like other approaches, this method has certain drawbacks. The major drawback is that the procedure is more complicated compared to the fuzzy or rough approach alone, since both approaches are

combined. Additionally, determining the limits of rough numbers with individual fuzzy numbers is challenging, with different methods providing different results. Each approach has its advantages and disadvantages. The advantages must be capitalized on and the shortcomings removed in future research. It should be noted that this methodology can be used in all other decision-making problems where expert opinion is involved. Under these circumstances, it is necessary to correct the criteria and alternatives that are used, while the steps and procedures remain the same.

This research revealed that electric vehicle manufacturers need to increase the range of their vehicles, speed up battery charging, and make their vehicles more competitive in terms of price. Only in this way will the expansion of these vehicles on the market take place and their use in sustainable urban logistics increase. Thus, the environment will be protected and the quality of life in urban areas will be improved. In order to solve this problem, it is necessary for countries to provide institutional support so that these vehicles are used as much as possible in practice.

## 7. Conclusions

Electric delivery vehicles are currently the best alternative to classic vehicles with internal combustion engines. In order to protect the environment and reduce delivery costs in urban areas, this paper discussed the introduction of electric vehicles by the Lombardija Company, Brčko. For this purpose, expert decision making and methods based on fuzzy–rough numbers were used. The reason for implementing these methods was the possibility of applying linguistic values and including uncertainty in decision making. Thus, decision making was adapted to the decision makers. According to the experts' estimates, the results showed that the range and price were key when buying electric delivery vehicles, and that the Citroen e-Berlingo XL (V1) vehicle best met the objectives of this research.

During the validation of the research results using other fuzzy–rough methods for vehicle ranking, it was established that all methods gave different rankings. Accordingly, it was shown that each MCDM method has its own characteristics and steps that determine the ranking of alternatives. Thus, when implementing the fuzzy–rough MABAC method, the best-ranked vehicle was the Renault Kangoo E-Tech Electric (V2). Due to these specificities, decision makers cannot rely on only one method, but must consider several methods in order to be more confident in their decisions. The validation of the results showed that the Citroën e-Berlingo was the best-ranked vehicle according to the five methods, representing the first choice for application in sustainable urban logistics by the Lombardija Company, Brčko. Sensitivity analysis showed that the vehicle was selected specifically because of its price. The sensitivity analysis also showed that the criterion weights had a very significant role in determining the ranking of vehicles.

This paper also revealed certain limitations regarding the implementation of this approach, which should be resolved in future research. First, the implementation of fuzzy–rough methods is much more complex for decision makers, so they must be acquainted with the basics of these methods. However, this could be solved by developing software programs that support decision making. With these programs, the decision maker would only need to evaluate criteria and alternatives, and the program itself would perform calculations. In addition, these programs must enable the implementation of several different analyses and methods that could provide more information for decision making. The specifics and steps of certain methods represent limitations for decision makers. Thus, by introducing program support, these actors would not have to understand how these methods work, but only the results they provide. Based on this, it is necessary to provide decision support programs that can facilitate decision making. When implementing these programs, it is necessary to ensure efficiency and scalability with good performance. The next limitation of this research was reflected in the selection of the vehicles themselves. New vehicles are introduced on the market every day, so it is necessary to include these vehicles in future research. Moreover, there are vehicles on the global market that would perhaps better meet the needs of the Lombardija Company, but they are not present on the

European market. Therefore, other vehicles that exist on the market of certain countries and for which there is an authorized service should be included in other similar studies.

This paper showed that methods based on fuzzy–rough numbers could be successfully used in future research and that the implementation of these methods allows one to exploit the advantages of both fuzzy sets and rough sets in decision making. In future research, it is necessary to develop new methods and approaches that can facilitate decision making and at the same time provide a certain level of confidence to the decision maker. By applying fuzzy–rough numbers, subjectivity is included in the decision-making process, while that subjectivity is reduced when making the final decision. In addition, uncertainty is included in decision making. Because of all this, a decision made using fuzzy–rough numbers is safer for the decision maker, so other approaches based on this methodology should be developed in future research.

In subsequent research, it is necessary to examine whether electric vehicles are more cost-effective than classic fossil fuel vehicles. For this purpose, it is necessary to consider the price of the vehicle, since electric vehicles are more expensive than classic vehicles. It is then necessary to compare the costs of battery charging and fueling in subsequent research. It should be noted that electric vehicles have recently experienced an expansion, so it is expected that they will progress in the future. Thus, in future research it is necessary to examine whether the development of these vehicles is sufficient to switch completely to these vehicles in the future. Furthermore, researchers should implement new approaches, such as Pythagorean fuzzy soft Einstein ordered intuitionistic fuzzy sets, to investigate how much these vehicles would reduce pollution and apply other methods of multi-criteria analysis such as TOPSIS, AHP, and VIKOR.

**Author Contributions:** Conceptualization, N.W., A.F.A. and Y.X.; methodology, A.P. and Ž.S.; validation, A.P. and A.F.A.; formal analysis, Ž.S.; data curation, A.P.; writing—original draft preparation, N.W. and Y.X.; writing—review and editing, A.P. and Ž.S.; visualization, N.W. and Y.X. All authors have read and agreed to the published version of the manuscript.

**Funding:** We acknowledge funding support from the National Social Science Fund of China (grant number 20BGL029, project name "The Two-way Governance Model and Dynamic Optimization Mechanism of Cross-border Mergers and Acquisitions of State-owned Enterprises in Chinese Mixed Ownership Reform").

**Institutional Review Board Statement:** Not applicable.

**Informed Consent Statement:** Not applicable.

**Data Availability Statement:** Not applicable.

**Acknowledgments:** The authors extend their appreciation to King Saud University for funding this work through the Researchers Supporting Project (number RSP2023R323), King Saud University, Riyadh, Saudi Arabia.

**Conflicts of Interest:** The authors declare no conflict of interest.

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
