# Peer review of "Multi-Criteria Selection of Electric Delivery Vehicles Using Fuzzy–Rough Methods"

_sustainability, doi:10.3390/su152115541_

Round 1

Reviewer 1 Report

Comments and Suggestions for Authors

Regarding this paper, the following should be considered:

1- This article presents a decision-making method for the specific selection context used. But what is the innovation of this article? Is it just a case study innovation of this article?

2- The research gap should be presented to distinguish this work from previous works.

3- The research methodology and method is not clear at all. Just presenting equations is not enough. The research method should be explained. Providing a graphic template can also be useful.

4- The results must be fully explained. I see now only a set of tables that need to be explained.

5- The results of this work should be compared with previous similar works.

6- The limitations of the research should be presented.

Author Response

Reviewer 1

Regarding this paper, the following should be considered:

Thank you for your constructive suggestions to make our paper as good as possible. We tried to incorporate all the suggestions into the paper.

  • This article presents a decision-making method for the specific selection context used. But what is the innovation of this article? Is it just a case study innovation of this article?

What is innovative in this paper is written in the abstract.

An innovative approach when selecting these vehicles is the application of a fuzzy-rough approach based on expert decision-making, where the decision-making process is adapted to decision-makers..

  • The research gap should be presented to distinguish this work from previous works.

The research gaps were already written in the second selection. In this way, we have added some more parts to the research gaps, namely:

These selections were based mainly on the technical characteristics of vehicles [27-30, 35-36], where linguistic evaluations were used to determine the importance of criteria [27, 29-31, 33], or objective methods to calculate the significance of criteria [35], while some papers used the combinations of technical characteristics and experts' linguistic assessments [39].

If in practice only the technical specifications of vehicles were considered, most users would have the same vehicle, which is not the case.

MARCOS method has not been used in research papers up to now, even though the MARCOS method has been used in several hundreds of papers.

  • The research methodology and method is not clear at all. Just presenting equations is not enough. The research method should be explained. Providing a graphic template can also be useful.

Methodologies were added to selection three.

Methodology and Methods

During the implementation of this research, the phases shown in Figure 1 were used.

Phase 1. Preparation of research phases

- Selection of experts

- Identification of limitations when selecting alternatives

- Selection of criteria

Phase 2. Evaluation of criteria and alternatives

- Evaluation of criteria by experts

- Evaluation of alternatives by experts

Phase 3. Preparation for analysis

- Transformation of linguistic estimates  into fuzzy numbers

- Application of rough approach

- Formation of an initial decision matrix

Phase 4. Conducting the analysis

- Determination of weights using the fuzzy-rough SWARA method

- Ranking of alternatives using the fuzz-rough MARCOS method

Phase 5. Confirmation of results

- Carrying out the validation of the results

- Carrying out sensitivity analysis

Figure 1. Research phases

When conducting this research, first, experts were selected and then they identified the limitations in the selection of alternatives and the criteria by which these alternatives would be observed. After that, the experts evaluated criteria and alternatives using linguistic values. In order to use these values when obtaining results, the values are first transformed into fuzzy numbers, and then the lower and upper limits of individual fuzzy numbers are determined and an initial decision matrix is formed. This matrix represents the basis for conducting analyses, i.e. for determining weights using the fuzzy-rough SWARA method and forming a ranking list using the fuzzy-rough MARCOS method. Finally, the obtained results will be confirmed by conducting the validation of the results and determining the importance of the criteria forming a ranking list of alternatives by conducting a sensitivity analysis.

  • The results must be fully explained. I see now only a set of tables that need to be explained.

It is explained with examples how the results were obtained, namely:

On an example of criterion C1, its value are formed as follows: . The values for other criteria are calculated in the same way.

For criterion C1, it is formed in the following way: . The values for other criteria are formed in the same way.

On an example of C1, it is as follows: . The weights for all criteria are calculated in the same way.

On an example of vehicle 1, criterion 1 normalization is calculated as follows. First, the minimum value of the criterion is calculated since this criterion is of the cost type. That is the value [(1.9, 3.5) (3.5, 4.5) (4.5, 6.2)]. The value of V1 is then divided from this value, and the normalization is calculated as follows: Normalization is calculated in the same way for other criteria and alternatives.

On an example of vehicle 1, criterion 1 is calculated as follows:

The weighted decision matrix for other alternatives and criteria is calculated in the same way.

On an example of vehicle 1, it is as follows: ,

  • The results of this work should be compared with previous similar works.

The obtained results were compared with similar papers.

In contrast, in the research by Štilić et al. [29], where electric taxi vehicles are investigated, the charging time, fast charging time and range are the most important, while the weight of the vehicle and the power of the vehicle are the most important in the research by Puška et al. [35]. Based on this, it can be concluded that the individual importance of criteria is determined depending on the purpose of electric vehicles.

  • The limitations of the research should be presented.

Some limits were written in the conclusion, so some more were added.

When implementing these programs, it is necessary to ensure efficiency and scalability with good performance. The next limitation of this research is reflected in the selection of the vehicles themselves. New vehicles are introduced on the market every day, so it is necessary to include these vehicles in future research. Moreover, there are vehicles on the global market that perhaps better meet the needs of the Lombardija Company, but they are not present on the European market. Therefore, other vehicles that exist on the market of a certain country and for which there is an authorized service will be included in other similar studies.

Reviewer 2 Report

Comments and Suggestions for Authors

This manuscript provides a multi-criteria selection of several electric minivans as electric delivery vehicles using fuzzy rough methods. The result shows a visual comparison of the criteria among available electric minivans for the customer to choose the most suitable electric minivan for them.

1. The title is suggested to be revised to "A multi-criteria selection of electric delivery vehicles using fuzzy rough methods".

2. In my opinion, what has not been discussed here is which criteria make them migrate from fossil-fueled vehicles to electric vehicles. For example, how comparable is the price of the two comparing vehicles? How comparable are the charging cost and fuel cost for daily usage? Perhaps, this issue can be added as future work in the conclusion section.

Comments on the Quality of English Language

-

Author Response

Reviewer 2

This manuscript provides a multi-criteria selection of several electric minivans as electric delivery vehicles using fuzzy rough methods. The result shows a visual comparison of the criteria among available electric minivans for the customer to choose the most suitable electric minivan for them.

Thank you for your constructive suggestions to make our paper as good as possible. We tried to incorporate all the suggestions into the paper.

  1. The title is suggested to be revised to "A multi-criteria selection of electric delivery vehicles using fuzzy rough methods".

The title has been corrected

  1. In my opinion, what has not been discussed here is which criteria make them migrate from fossil-fueled vehicles to electric vehicles. For example, how comparable is the price of the two comparing vehicles? How comparable are the charging cost and fuel cost for daily usage? Perhaps, this issue can be added as future work in the conclusion section.

This was added in the guidelines for future research and also mentioned in the methodology.

In subsequent research, it is necessary to examine whether electric vehicles are more cost-effective than classic fossil fuel vehicles. For this purpose, it is necessary to include the price of the vehicle since electric vehicles are more expensive than classic vehicles. It is then necessary to compare the costs of battery charging and fueling in subsequent research. It should be noted that electric vehicles have recently experienced an expansion, so it is expected that they will progress in the future.

In order to use these vehicles for delivery, it is necessary to take into consideration several practical challenges and obstacles, some of which are as follows. Electric vehicles have not yet been accepted enough in practice, and there is a lack of fast charging stations for these vehicles. In addition, their price is also a limiting factor that must be considered when purchasing these vehicles. That is why this company will use a classic electrical network and charge the batteries using slow chargers, at least that is how it is planned to be. This is a problem because of the time it takes to charge a battery this way. Therefore, it is necessary to charge these vehicles only at night when they are not in use. For that reason, this company is considering the possibility of introducing these vehicles, which does not mean that they will be introduced. This decision would be influenced by the state's decision to introduce these vehicles. Then, a limitation when introducing these vehicles can be the delivery time of these vehicles because the Lombardija Company is here a customer, so the suppliers of electric delivery vehicles must adapt to it. 

Reviewer 3 Report

Comments and Suggestions for Authors

In this manuscript, the selection of electric vehicles that can be used as delivery vehicles has been made. Thus, it has been investigated which vehicle has the best characteristics to be used in sustainable urban logistics. When conducting the research, expert decision-making based on MCDM (SWARA and MARCOS in fuzzy-rough form) was applied. However, I have few observations:

1.      Improvement in the selection performance has been a traditional goal as far as MCDM are concerned. There are already a lot of decision making methods available in the literature. Authors are required to elaborate more clearly on the purpose and novelty of their work. Why only SWARA and MARCOS methods, why not any other like Fuzzy AHP?

22. Authors are required to write more clearly the scientific contribution, main limitation and future direction of their manuscript.

3.      Authors are required to go through recent references related to various recently developed in MCDA and their applications in the various fields to make the reference list exhaustive. For example:

(i) Rawat, S. S., et al. (2022). A state-of-the-art survey on analytical hierarchy process applications in sustainable development. Int. J. Math. Eng. Manag. Serv7, 883-917.

(ii) Pant, S., et al. (2023). AHP-based multi-criteria decision-making approach for monitoring health management practices in smart healthcare system. International Journal of System Assurance Engineering and Management, 1-12.

(iii) Kumar, A., & Pant, S. (2022). Analytical hierarchy process for sustainable agriculture: An overview. MethodsX, 101954. Kindly recheck all the captions of figures and tables.

4. Proofread the manuscript carefully for grammatical errors.

5. Abstract is too long keep it precise and understandable. 

6. Recheck all the captions of figures, equations and tables.

Comments on the Quality of English Language

Proofread the manuscript carefully for grammatical errors.

Author Response

Reviewer 3

In this manuscript, the selection of electric vehicles that can be used as delivery vehicles has been made. Thus, it has been investigated which vehicle has the best characteristics to be used in sustainable urban logistics. When conducting the research, expert decision-making based on MCDM (SWARA and MARCOS in fuzzy-rough form) was applied. However, I have few observations:

Thank you for your constructive suggestions to make our paper as good as possible. We tried to incorporate all the suggestions into the paper.

  1. Improvement in the selection performance has been a traditional goal as far as MCDM are concerned. There are already a lot of decision making methods available in the literature. Authors are required to elaborate more clearly on the purpose and novelty of their work. Why only SWARA and MARCOS methods, why not any other like Fuzzy AHP?

It was explained that the SWARA and MARCOS methods were used

The purpose of this method is for decision-makers to first rank criteria according to their importance, and then to determine their weights by comparing those criteria. Unlike the AHP method, where each criterion must be compared with each other [47], this does not have to be done with the SWARA method. Ranking and then the evaluation of the criteria based on the ranking are performed. In this way, it is not necessary to compare each individual criterion, but only the worse ranked compared to the better ranked. Thus, the number of comparisons is reduced to n-1 [48] in contrast to AHP where the number of comparisons is n·(n-1) [49]. The problem with the application of the AHP method is the number of criteria that are compared [50]. If there are a large number of criteria, it is difficult to achieve consistency in decision-making.

This method is one of the recent methods of multi-criteria analysis and as such has been used in many studies in a short period of time and has been accepted by various authors. Besides, its results do not deviate from the results of other methods and have been confirmed in hundreds of research papers in which it was used. These are just some of the reasons why this method has been used here. Moreover, it has not been used in fuzzy-rough form until now. The purpose of this method is to rank alternatives in relation to the ideal and anti-ideal solution in a way that an alternative that is closer to the ideal and further from the anti-ideal solution is better.

  1. Authors are required to write more clearly the scientific contribution, main limitation and future direction of their manuscript.

In the conclusion, the limits of the research were explained, but more limits were added throughout the paper. Some of these segments are as follows:

In order to use these vehicles for delivery, it is necessary to take into consideration several practical challenges and obstacles, some of which are as follows. Electric vehicles have not yet been accepted enough in practice, and there is a lack of fast charging stations for these vehicles. In addition, their price is also a limiting factor that must be considered when purchasing these vehicles. That is why this company will use a classic electrical network and charge the batteries using slow chargers, at least that is how it is planned to be. This is a problem because of the time it takes to charge a battery this way. Therefore, it is necessary to charge these vehicles only at night when they are not in use. For that reason, this company is considering the possibility of introducing these vehicles, which does not mean that they will be introduced. This decision would be influenced by the state's decision to introduce these vehicles. Then, a limitation when introducing these vehicles can be the delivery time of these vehicles because the Lombardija Company is here a customer, so the suppliers of electric delivery vehicles must adapt to it. 

When implementing these programs, it is necessary to ensure efficiency and scalability with good performance. The next limitation of this research is reflected in the selection of the vehicles themselves. New vehicles are introduced on the market every day, so it is necessary to include these vehicles in future research. Moreover, there are vehicles on the global market that perhaps better meet the needs of the Lombardija Company, but they are not present on the European market. Therefore, other vehicles that exist on the market of a certain country and for which there is an authorized service will be included in other similar studies.

However, electric delivery vehicles have certain limitations that slow down the introduction of these vehicles. The main limitations are related to the range, battery capacity, charging time and price of these vehicles. Due to these limitations, decision-makers in companies still opt for classic delivery vehicles. However, electric vehicles do not emit harmful gases into the atmosphere, unlike classic internal combustion vehicles [9]. The use of electric vehicles in urban logistics would reduce the impact on the environment. In order to reduce this impact, the energy used for charging electric vehicles must be produced from sustainable energy sources.

  1. Authors are required to go through recent references related to various recently developed in MCDA and their applications in the various fields to make the reference list exhaustive. For example:

(i) Rawat, S. S., et al. (2022). A state-of-the-art survey on analytical hierarchy process applications in sustainable development. Int. J. Math. Eng. Manag. Serv7, 883-917.

(ii) Pant, S., et al. (2023). AHP-based multi-criteria decision-making approach for monitoring health management practices in smart healthcare system. International Journal of System Assurance Engineering and Management, 1-12.

(iii) Kumar, A., & Pant, S. (2022). Analytical hierarchy process for sustainable agriculture: An overview. MethodsX, 101954. Kindly recheck all the captions of figures and tables.

These references were dropped in the paper when discussing the differences between the SWARA and AHP methods.

  1. Rawat, S. S., Pant, S., Kumar, A., Ram, M., Sharma, H. K., & Kumar, A. (2022). A state-of-the-art survey on analytical Hierarchy Process applications in sustainable development. International Journal of Mathematical, Engineering and Management Sciences, 7(6), 883–917. https://doi.org/10.33889/ijmems.2022.7.6.056
  2. Pant, S., Garg, P., Kumar, A., Ram, M., Kumar, A., Sharma, H. K., & Klochkov, Y. (2023). AHP-based multi-criteria decision-making approach for monitoring health management practices in smart healthcare system. International Journal of System Assurance Engineering and Management. Article in press. https://doi.org/10.1007/s13198-023-01904-5
  3. Kumar, A., & Pant, S. (2023). Analytical hierarchy process for sustainable agriculture: An overview. MethodsX, 10, 101954. https://doi.org/10.1016/j.mex.2022.101954
  4. Proofread the manuscript carefully for grammatical errors.

The paper has been sent for additional proofreading.

  1. Abstract is too long keep it precise and understandable. 

The abstract has been shortened and corrected. Corrections are marked in red font in the abstract.

  1. Recheck all the captions of figures, equations and tables.

Titles of tables and figures have been checked and corrected to make it clearer for readers.

Reviewer 4 Report

Comments and Suggestions for Authors

The topic of the paper has been discussed in many previous literature. The authors should try to highlight why these kinds of research are necessary with more illustrations and demonstrations.  The paper suffers from the following weaknesses:

(1) Although the paper mentions the use of a combined fuzzy-rough approach based on SWARA and MARCOS methods, it does not provide sufficient details about the application and suitability of these methods for the selection of electric delivery vehicles.

(2) The paper briefly mentions the use of electric delivery vehicles as the best solution for sustainable urban logistics without providing a comprehensive justification for this choice. A more in-depth analysis of the environmental and economic benefits, as well as the feasibility and limitations of electric vehicles, would have strengthened the argument.

(3) The paper does not adequately consider the perspectives of various stakeholders involved in sustainable urban logistics, such as logistics providers, policymakers, and customers. Incorporating stakeholder perspectives would have provided a more comprehensive understanding of the challenges and opportunities in implementing electric delivery vehicles. The profiles of the selected experts are not provided. 

(4) The paper does not discuss the generalizability of the proposed fuzzy-rough approach beyond the specific context of electric delivery vehicle selection. Considering the broader applicability and potential extensions of the methodology would have enhanced the practical value of the study.

(5) The paper states that the fuzzy-rough approach brings the selection of electric delivery vehicles closer to human thinking but does not explain how or provide evidence to support this claim. A more thorough analysis of the human decision-making process and the alignment with the proposed approach would have strengthened the argument.

(6) The paper does not adequately discuss the practical challenges and barriers that may arise during the implementation of electric delivery vehicles in urban logistics. Considering factors such as infrastructure limitations, regulatory requirements, and operational considerations would have provided a more realistic assessment of the feasibility and potential obstacles.

(7) The paper does not sufficiently address the importance of customer needs and preferences in the selection of electric delivery vehicles. Understanding factors such as delivery time windows, customization requirements, and service quality from the customer's perspective would have enhanced the relevance and practicality of the methodology.

(8) The paper does not discuss the scalability of the proposed fuzzy-rough approach to accommodate larger-scale decision-making problems or future expansion of urban logistics. Considering the scalability and computational efficiency of the methodology would have provided insights into its applicability in real-world scenarios.

Comments on the Quality of English Language

The language quality of the paper could be improved. 

Author Response

Reviewer 4

The topic of the paper has been discussed in many previous literature. The authors should try to highlight why these kinds of research are necessary with more illustrations and demonstrations.  The paper suffers from the following weaknesses:

Thank you for your constructive suggestions to make our paper as good as possible. We tried to incorporate all the suggestions into the paper.

  • Although the paper mentions the use of a combined fuzzy-rough approach based on SWARA and MARCOS methods, it does not provide sufficient details about the application and suitability of these methods for the selection of electric delivery vehicles.

It was explained that the SWARA and MARCOS methods were used and that the fuzzy-rough approach was used.

Applying the fuzzy-rough approach in decision-making, specifically in the selection of electric delivery vehicles, the entire decision-making process is adjusted to users' requirements. Thus, the decision is not only influenced by the technical characteristics of these vehicles, but also including users' preferences, which is what is done by applying this approach. In addition, the combination of fuzzy and rough approaches first enables linguistic values to be used in decision-making, which is more suitable for decision-makers. Then, the fuzzy approach enables these linguistic values to be used to obtain final results, while the application of the rough approach in this decision-making includes uncertainty in the decision-making process. By applying the fuzzy-rough approach, all the advantages of both fuzzy and rough approach are used, and thus two approaches are mutually complemented. That is why this approach is better than classic fuzzy and rough approaches in decision-making [18].    

  • The paper briefly mentions the use of electric delivery vehicles as the best solution for sustainable urban logistics without providing a comprehensive justification for this choice. A more in-depth analysis of the environmental and economic benefits, as well as the feasibility and limitations of electric vehicles, would have strengthened the argument.

It was additionally explained that it is necessary to use electric vehicles for distribution in urban logistics.

However, electric delivery vehicles have certain limitations that slow down the introduction of these vehicles. The main limitations are related to the range, battery capacity, charging time and price of these vehicles. Due to these limitations, decision-makers in companies still opt for classic delivery vehicles. However, electric vehicles do not emit harmful gases into the atmosphere, unlike classic internal combustion vehicles [9]. The use of electric vehicles in urban logistics would reduce the impact on the environment. In order to reduce this impact, the energy used for charging electric vehicles must be produced from sustainable energy sources. Based on that, the motivation for this paper is to apply sustainability in urban logistics using electric delivery vehicles

  • The paper does not adequately consider the perspectives of various stakeholders involved in sustainable urban logistics, such as logistics providers, policymakers, and customers. Incorporating stakeholder perspectives would have provided a more comprehensive understanding of the challenges and opportunities in implementing electric delivery vehicles. The profiles of the selected experts are not provided. 

In this paper, it is observed from the point of view of the company Lombardia, which appears as a possible buyer of electric vehicles. So the whole process is viewed from the point of view of the company that is the customer. It was additionally explained which experts were chosen.

The first expert from the company is an economics graduate with many years of experience in logistics business who communicates with customers on a daily basis. Thus, the expert shared his knowledge based on experience with customers for the purposes of this research. The other two experts are drivers who carry out distribution. They are familiar with whom distribution is carried out and they have many years of experience with vehicles.

  • The paper does not discuss the generalizability of the proposed fuzzy-rough approach beyond the specific context of electric delivery vehicle selection. Considering the broader applicability and potential extensions of the methodology would have enhanced the practical value of the study.

In the guidelines for future research, it is explained that the mentioned methodology can be used in some other areas, but the criteria and alternatives must be changed.

The methodology used in this research is recent and as such must be compared with other approaches used by other methods. The basis of decision-making is that the decision-maker has all the information regarding the decision, so then it is necessary to apply different methods and procedures. This paper has provided a way in which the fuzzy-rough approach can be used in decision making. In further methods, the approach needs to be compared with other approaches in order to be able to establish what are its advantages and what are its disadvantages. Some of the advantages of this approach are the combination of fuzzy and rough approaches where the advantages of these methods are used, then in the development of a new approach when using the MARCOS method, and then including uncertainty in decision-making. However, like other approaches, this one has certain drawbacks. The major drawback is a more complicated procedure compared to the fuzzy or rough approach itself, since both approaches are combined. Then, the problem is when determining the limits of rough numbers with individual fuzzy numbers, and the fact that different methods provide different results. Each approach has its advantages and disadvantages. The advantages must be used and the shortcomings removed in future research. When referring to the methodology, it should be noted that it can be used in all other decision-making problems where expert opinion is involved. Under these circumstances, it is necessary to correct the criteria and alternatives that were used, while the steps and procedures remain the same.

  • The paper states that the fuzzy-rough approach brings the selection of electric delivery vehicles closer to human thinking but does not explain how or provide evidence to support this claim. A more thorough analysis of the human decision-making process and the alignment with the proposed approach would have strengthened the argument.

It was explained why linguistic values were used and why they are closer to human thinking.

The purpose of this methodology is to bring decision-making closer to the deci-sion-maker. The decision-maker does not have to provide exact information, but it is possible to use imprecise information in a form of linguistic values [40]. These values make it possible to value not only qualitative criteria but also quantitative ones [39]. In that case, estimates are given in a form of good, bad, medium, etc. It is easier for people to say that something is good or very good or excellent than to evaluate it with numerical ratings using, e.g., a grade 4 or 5 [41]. In this way, by using these values, this research is more adapted to human thinking.

(6) The paper does not adequately discuss the practical challenges and barriers that may arise during the implementation of electric delivery vehicles in urban logistics. Considering factors such as infrastructure limitations, regulatory requirements, and operational considerations would have provided a more realistic assessment of the feasibility and potential obstacles.

Throughout the paper, the shortcomings of electric vehicles were included in the introduction, methodology, discussion and conclusion. The biggest limitations mentioned in this paper are price, range, battery capacity and charging.

(7) The paper does not sufficiently address the importance of customer needs and preferences in the selection of electric delivery vehicles. Understanding factors such as delivery time windows, customization requirements, and service quality from the customer's perspective would have enhanced the relevance and practicality of the methodology.

The company Lombardia is a potential buyer of these vehicles and the whole paper looks at the decision-making about the purchase of these vehicles through its perspective as a buyer.

(8) The paper does not discuss the scalability of the proposed fuzzy-rough approach to accommodate larger-scale decision-making problems or future expansion of urban logistics. Considering the scalability and computational efficiency of the methodology would have provided insights into its applicability in real-world scenarios.

When the program support was mentioned in the conclusion, it was added that it must be efficient and scalable with good performance

When implementing these programs, it is necessary to ensure efficiency and scalability with good performance.

Reviewer 5 Report

Comments and Suggestions for Authors

Title: Application of a decision-making model under uncertainty for selecting electric delivery vehicles

Authors used the multi criteria decision making problem for the selection of electric vehicles that can be used as delivery vehicles has been made. authors investigated which vehicle has the best characteristics to be used in sustainable urban logistics. For this authors used the Stepwise Weight Assessment Ratio Analysis and Measurement Alternatives and Ranking according to Compromise Solution methods in fuzzy-rough form were used. The presented idea is interesting and suitable for “Sustainability”. But, some necessary changes required before its acceptance given as follows::

1. The authors need to revise the abstract in a precise manner because abstract is too much leangthy. 2. The introduction should be revised. It does not show the research gap, contribution and objectives. The authors have to explain it in a wide manner. Also some recent studies about sustainable supplier selection and MCDM are missing. Authors should revise their introduction according to these studies. (a): Pythagorean fuzzy soft Einstein ordered weighted average operator in sustainable supplier selection problem. (b): sustainable practices to reduce environmental impact of industry using interaction aggregation operators under interval-valued pythagorean fuzzy hypersoft set. (c): selection of best alternative for an automotive company by intuitionistic fuzzy topsis method. (d): selection of medical clinic for disease diagnosis by using topsis method 3. Some sentences in the text need to be rewritten. It should be noted that the manuscript needs to be carefully edited by professional English editors, paying special attention to English grammar, spelling, and sentence structure so that readers can clearly understand the objectives and results of the research. 4. Indicate the challenges of existing literature in the selection of an electric delivery vehicle decisions. 5. The motivation of current research is not enough. The authors need to discuss this appropriately. 6. Authors must compare their proposed method and decision model with other recent studies and express its advantages and benefits. 7. The conclusion is too long and with several useless sentences. Please revise the conclusion in a precise manner. Comments on the Quality of English Language

Some sentences in the text need to be rewritten. It should be noted that the manuscript needs to be carefully edited by professional English editors, paying special attention to English grammar, spelling, and sentence structure so that readers can clearly understand the objectives and results of the research.

Author Response

Reviewer 5

Title: Application of a decision-making model under uncertainty for selecting electric delivery vehicles

Thank you for your constructive suggestions to make our paper as good as possible. We tried to incorporate all the suggestions into the paper.

Authors used the multi criteria decision making problem for the selection of electric vehicles that can be used as delivery vehicles has been made. authors investigated which vehicle has the best characteristics to be used in sustainable urban logistics. For this authors used the Stepwise Weight Assessment Ratio Analysis and Measurement Alternatives and Ranking according to Compromise Solution methods in fuzzy-rough form were used. The presented idea is interesting and suitable for “Sustainability”. But, some necessary changes required before its acceptance given as follows::

  1. The authors need to revise the abstract in a precise manner because abstract is too much leangthy.

The abstract was corrected, shortened and refined. All changes in the abstract are marked in red font.

  1. The introduction should be revised. It does not show the research gap, contribution and objectives. The authors have to explain it in a wide manner. Also some recent studies about sustainable supplier selection and MCDM are missing. Authors should revise their introduction according to these studies. (a): Pythagorean fuzzy soft Einstein ordered weighted average operator in sustainable supplier selection problem. (b): sustainable practices to reduce environmental impact of industry using interaction aggregation operators under interval-valued pythagorean fuzzy hypersoft set. (c): selection of best alternative for an automotive company by intuitionistic fuzzy topsis method. (d): selection of medical clinic for disease diagnosis by using topsis method

Research gaps are found at the end of the literature review selection. After that, the choice of supplier has nothing to do with the topic of this research, because the topic is the choice of an electric delivery vehicle and not the choice of a supplier of electric delivery vehicles. We put that in the guidelines for future research.

  1. Some sentences in the text need to be rewritten. It should be noted that the manuscript needs to be carefully edited by professional English editors, paying special attention to English grammar, spelling, and sentence structure so that readers can clearly understand the objectives and results of the research.

The research objectives were corrected and the entire paper was proofread.

Based on this, the contribution of this paper is reflected in the following:

  • Improving sustainable urban logistics using electric delivery vehicles.
  • Applying innovative methodology for the selection of an electric delivery vehicle based on a fuzzy-rough approach.
  • Using a fuzzy-rough approach when selecting electric delivery vehicles, adapting decision-making process to human preferences.
  • Selecting an electric vehicle that best meets the sustainability goals of urban logistics.
  • Promoting the use of electric delivery vehicles in urban logistics application.
  1. Indicate the challenges of existing literature in the selection of an electric delivery vehicle decisions.

In the conclusion, we added these challenges, and what can be used in future research.

In subsequent research, it is necessary to examine whether electric vehicles are more cost-effective than classic fossil fuel vehicles. For this purpose, it is necessary to include the price of the vehicle since electric vehicles are more expensive than classic vehicles. It is then necessary to compare the costs of battery charging and fueling in subsequent research. It should be noted that electric vehicles have recently experienced an expansion, so it is expected that they will progress in the future. Thus, in future research it is necessary to examine whether the development of these vehicles is sufficient to switch completely to these vehicles in the future. In future research, it is necessary to use some new approaches such as the use of Pythagorean fuzzy soft Einstein ordered, intuitionistic fuzzy set, then to investigate how much these vehicles would reduce pollution, and also to apply some other methods of multi-criteria analysis such as TOPSIS, AHP, VIKOR and various other methods.

  1. The motivation of current research is not enough. The authors need to discuss this appropriately.

The motivation of the introduction of the paper was added to the introduction.

Based on that, the motivation for this paper is to apply sustainability in urban logistics using electric delivery vehicles.    

  1. Authors must compare their proposed method and decision model with other recent studies and express its advantages and benefits.

This was done in the discussion.

Through the validation of the results in this research, in addition to comparing the results of different methods, those methods were also compared. The results showed that different methods provide different results precisely because of the steps they apply. The greatest difference was shown by the MABAC method, and this is due to the fact that this method uses different data normalization and the ranking is based on the average value of alternatives. In addition, the weighting process is different compared to other methods.

The methodology used in this research is recent and as such must be compared with other approaches used by other methods. The basis of decision-making is that the decision-maker has all the information regarding the decision, so then it is necessary to apply different methods and procedures. This paper has provided a way in which the fuzzy-rough approach can be used in decision making. In further methods, the approach needs to be compared with other approaches in order to be able to establish what are its advantages and what are its disadvantages. Some of the advantages of this approach are the combination of fuzzy and rough approaches where the advantages of these methods are used, then in the development of a new approach when using the MARCOS method, and then including uncertainty in decision-making. However, like other approaches, this one has certain drawbacks. The major drawback is a more complicated procedure compared to the fuzzy or rough approach itself, since both approaches are combined. Then, the problem is when determining the limits of rough numbers with individual fuzzy numbers, and the fact that different methods provide different results. Each approach has its advantages and disadvantages. The advantages must be used and the shortcomings removed in future research. When referring to the methodology, it should be noted that it can be used in all other decision-making problems where expert opinion is involved. Under these circumstances, it is necessary to correct the criteria and alternatives that were used, while the steps and procedures remain the same.

  1. The conclusion is too long and with several useless sentences. Please revise the conclusion in a precise manner.

We specified the conclusion. We removed a few sentences from it, but due to reviewers' requests, we had to add limits and guidelines for future research.

Reviewer 6 Report

Comments and Suggestions for Authors

The research topic is meaningful and interesting, but the authors should clear indicate the main contributions of this study.

1. Abstract. The content is lengthy, the focus is not prominent, and the innovation is insufficient.

2. Introduction. (1) For the background part of the research, the length is long without highlighting the key points. (2) The description of the advantages of the fuzzy-rough method and the disadvantages of the existing methods is insufficient. (3) Insufficient expression of research contribution, over-emphasis on what I have done, and no specific explanation of my innovation points.

3. Methods. This part of the content is the application of existing methods, and the description of innovative content is insufficient.

4. Case study. The evaluation indicators in Table 1 should be the focus of this study, but there is a lack of necessary description of the process of indicator screening. Why are there 10 indicators instead of 8?

5. Conclusion. Environmental protection is mentioned in the whole paper, but there is no effective data or theoretical support in the methods and conclusions.

Author Response

Reviewer 6

The research topic is meaningful and interesting, but the authors should clear indicate the main contributions of this study.

Thank you for your constructive suggestions to make our paper as good as possible. We tried to incorporate all the suggestions into the paper.

  1. The content is lengthy, the focus is not prominent, and the innovation is insufficient.

The abstract was corrected, and the innovation of this paper was added to it. All changes in the abstract are marked in red font.

An innovative approach when selecting these vehicles is the application of a fuzzy-rough approach based on expert decision-making, where the decision-making process is adapted to decision-makers.

  1. (1) For the background part of the research, the length is long without highlighting the key points. (2) The description of the advantages of the fuzzy-rough method and the disadvantages of the existing methods is insufficient. (3) Insufficient expression of research contribution, over-emphasis on what I have done, and no specific explanation of my innovation points.

The introduction has been additionally corrected, all changes are marked in red font. In it, we tried to incorporate all these suggestions of you and other reviewers.

However, electric delivery vehicles have certain limitations that slow down the introduction of these vehicles. The main limitations are related to the range, battery capacity, charging time and price of these vehicles. Due to these limitations, decision-makers in companies still opt for classic delivery vehicles. However, electric vehicles do not emit harmful gases into the atmosphere, unlike classic internal combustion vehicles [9]. The use of electric vehicles in urban logistics would reduce the impact on the environment. In order to reduce this impact, the energy used for charging electric vehicles must be produced from sustainable energy sources. Based on that, the motivation for this paper is to apply sustainability in urban logistics using electric delivery vehicles.  

This paper tends to promote the use of electric vehicles in the distribution of goods using examples from practice. The aim of this paper is to select an electric delivery vehicle that would best meet the sustainability goals of urban logistics for the distribution of goods. Electric delivery vehicles have similar technical characteristics and the selection of these vehicles cannot be based only on these technical features. In this research, a combined fuzzy-rough approach will be applied in the selection of electric delivery vehicles based on fuzzy-rough SWARA and MARCOS methods. Applying the fuzzy-rough approach in decision-making, specifically in the selection of electric delivery vehicles, the entire decision-making process is adjusted to users' requirements. Thus, the decision is not only influenced by the technical characteristics of these vehicles, but also including users' preferences, which is what is done by applying this approach. In addition, the combination of fuzzy and rough approaches first enables linguistic values to be used in decision-making, which is more suitable for decision-makers. Then, the fuzzy approach enables these linguistic values to be used to obtain final results, while the application of the rough approach in this decision-making includes uncertainty in the decision-making process. By applying the fuzzy-rough approach, all the advantages of both fuzzy and rough approach are used, and thus two approaches are mutually complemented. That is why this approach is better than classic fuzzy and rough approaches in decision-making [18].    

Based on this, the contribution of this paper is reflected in the following:

  • Improving sustainable urban logistics using electric delivery vehicles.
  • Applying innovative methodology for the selection of an electric delivery vehicle based on a fuzzy-rough approach.
  • Using a fuzzy-rough approach when selecting electric delivery vehicles, adapting decision-making process to human preferences.
  • Selecting an electric vehicle that best meets the sustainability goals of urban logistics.
  • Promoting the use of electric delivery vehicles in urban logistics application.
  1. This part of the content is the application of existing methods, and the description of innovative content is insufficient.

The methods and why they were used and why the fuzzy-rough approach was used were additionally explained.

The purpose of this methodology is to bring decision-making closer to the decision-maker. The decision-maker does not have to provide exact information, but it is possible to use imprecise information in a form of linguistic values [40]. These values make it possible to value not only qualitative criteria but also quantitative ones [39]. In that case, estimates are given in a form of good, bad, medium, etc. It is easier for people to say that something is good or very good or excellent than to evaluate it with numerical ratings using, e.g., a grade 4 or 5 [41]. In this way, by using these values, this research is more adapted to human thinking.

The purpose of this method is for decision-makers to first rank criteria according to their importance, and then to determine their weights by comparing those criteria. Unlike the AHP method, where each criterion must be compared with each other [47], this does not have to be done with the SWARA method. Ranking and then the evaluation of the criteria based on the ranking are performed. In this way, it is not necessary to compare each individual criterion, but only the worse ranked compared to the better ranked. Thus, the number of comparisons is reduced to n-1 [48] in contrast to AHP where the number of comparisons is n·(n-1) [49]. The problem with the application of the AHP method is the number of criteria that are compared [50]. If there are a large number of criteria, it is difficult to achieve consistency in decision-making. Anyway, that is another reason why the SWARA method will be used.

This method is one of the recent methods of multi-criteria analysis and as such has been used in many studies in a short period of time and has been accepted by various authors. Besides, its results do not deviate from the results of other methods and have been confirmed in hundreds of research papers in which it was used. These are just some of the reasons why this method has been used here. Moreover, it has not been used in fuzzy-rough form until now. The purpose of this method is to rank alternatives in relation to the ideal and anti-ideal solution in a way that an alternative that is closer to the ideal and further from the anti-ideal solution is better.

  1. Case study. The evaluation indicators in Table 1 should be the focus of this study, but there is a lack of necessary description of the process of indicator screening. Why are there 10 indicators instead of 8?

The criteria in this research are additionally explained.

Electric delivery vehicles are more expensive than classic delivery vehicles, so the price was taken as a criterion. For the observed company, it is important to buy a vehicle that is affordable and to satisfy most of the goals. Certainly, vehicles that are more expensive have more equipment included. Acceleration is important for delivery vehicles as it is necessary to deliver goods to various places. These vehicles are battery limited and their range depends on it. If the battery is larger, the vehicle is more expensive, but at the same time it has a larger range. The power of the engine is also important because it is not the same to drive an empty or a full vehicle. In order to transport a large amount of goods, the vehicle needs to have a more powerful engine. Battery capacity is important for the range of electric delivery vehicles. The larger the battery capacity, the greater the range of the vehicle. However, this negatively affects the charging of those batteries. If the capacity is higher, the time is also longer, and it is desirable to reduce the charging time because it is necessary to deliver goods on time. That is why fast chargers are used so batteries can be charged faster. As it is for classic vehicles where fuel consumption is important, vehicle consumption is equivalent for electric vehicles. It is necessary that the consumption is lower in order to increase the range of the vehicle. In the case of delivery vehicles, the load capacity of these vehicles is significant, too. It is desirable that the load capacity is as high as possible. Also, the same applies to the capacity of the trunk, which should be large.

In order to use these vehicles for delivery, it is necessary to take into consideration several practical challenges and obstacles, some of which are as follows. Electric vehicles have not yet been accepted enough in practice, and there is a lack of fast charging stations for these vehicles. In addition, their price is also a limiting factor that must be considered when purchasing these vehicles. That is why this company will use a classic electrical network and charge the batteries using slow chargers, at least that is how it is planned to be. This is a problem because of the time it takes to charge a battery this way. Therefore, it is necessary to charge these vehicles only at night when they are not in use. For that reason, this company is considering the possibility of introducing these vehicles, which does not mean that they will be introduced. This decision would be influenced by the state's decision to introduce these vehicles. Then, a limitation when introducing these vehicles can be the delivery time of these vehicles because the Lombardija Company is here a customer, so the suppliers of electric delivery vehicles must adapt to it.

  1. Conclusion. Environmental protection is mentioned in the whole paper, but there is no effective data or theoretical support in the methods and conclusions.

We specified the conclusion. We removed a few sentences from it, but due to reviewers' requests, we had to add limits and guidelines for future research.

Round 2

Reviewer 1 Report

Comments and Suggestions for Authors

The authors have made the necessary corrections and the article is acceptable in its current state.

Best Regards

Reviewer 3 Report

Comments and Suggestions for Authors

Suggested corrections implemented.

Comments on the Quality of English Language

Minor editing required 

Reviewer 4 Report

Comments and Suggestions for Authors

I have no further comments to contribute.

Reviewer 5 Report

Comments and Suggestions for Authors

The authors revised the manuscript according to my suggestions.

Reviewer 6 Report

Comments and Suggestions for Authors

This manuscript has been revised in detailed according to my opinion and agreed to be accepted!